# Subglacial drainage patterns of Devon Island, Canada: Detailed comparison of rivers and subglacial meltwater channels

Anna Grau Galofre[1], A. Mark Jellinek[1], Gordon R. Osinski[2], Michael Zanetti[3], and Antero Kukko[4]

[1]Department of Earth, Ocean and Atmospheric Sciences, The University of British Columbia.
[2]Department of Physics and Astronomy, University of Western Ontario
[3]Department of Earth Sciences and Centre for Planetary Science and Exploration, University of Western Ontario
[4]Center of Excellence in Laser Scanning Research, Department of Remote Sensing and Photogrammetry, Finnish Geospatial Research Institute

*Correspondence to:* Anna Grau Galofre (agraugal@eos.ubc.ca)

**Abstract.**

Subglacial meltwater channels (N-channels) are attributed to erosion by meltwater in subglacial conduits. They exert a major control on meltwater accumulation at the base of ice sheets, serving as drainage pathways and modifying ice flow rates. The study of exposed relict subglacial channels offers a unique opportunity to characterize the geomorphologic fingerprint of subglacial erosion as well as study the structure and characteristics of ice sheet drainage systems. In this study we present detailed field and remote sensing observations of exposed subglacial meltwater channels in excellent preservation state on Devon Island (Canadian Arctic Archipelago). We characterize channel cross section, longitudinal profiles, and network morphologies and establish the spatial extent and distinctive characteristics of subglacial drainage systems. We use field-based GPS measurements of subglacial channel longitudinal profiles, along with stereo imagery derived Digital Surface Models (DSM), and novel kinematic portable LiDAR data to establish a detailed characterization of subglacial channels in our field study area, including their distinction from rivers and other meltwater drainage systems. Subglacial channels typically cluster in groups of ∼ 10 channels and are oriented perpendicular to active or former ice margins. Although their overall direction generally follows topographic gradients, channels can be oblique to topographic gradients and have undulating longitudinal profiles. We also observe that the width of first order tributaries is one to two orders of magnitude larger than in Devon Island river systems, and approximately constant. Furthermore, our findings are consistent with theoretical expectations drawn from analyses of flow driven by gradients in effective water pressure related to variations in ice thickness. Our field and remote sensing observations represent the first high resolution study of the subglacial geomorphology of the high Arctic, and provide quantitative and qualitative descriptions of subglacial channels that revisit well-established field identification guidelines. Distinguishing subglacial channels in topographic data is critical for understanding the emergence, geometry and extent of channelized meltwater systems and their role in ice sheet drainage. The final aim of this study is to facilitate the identification of subglacial channel networks throughout the globe by using remote sensing techniques, which will improve the detection of these systems and help to build understanding of the underlying mechanics of subglacial channelized drainage.

# 1 Introduction

Subglacial meltwater channels, often referred to as N-channels, are the erosional expression of turbulent flows in pressurized subglacial channels. Together with subglacial channels incised in overlying ice (R-channels), they modulate meltwater accumulation at the base of ice sheets and serve as highly efficient drainage pathways carrying meltwater to the ice terminus (e.g., Röthlisberger, 1972; Weertman, 1972; Nye, 1976; Sugden et al., 1991; Greenwood et al., 2007; Kehew et al., 2012). In particular, the transition from distributed to channelized drainage leads to a reduction in ice flow rates, modifying ice loss rates and enhancing surging (e.g., Schoof, 2010). Subglacial channelized drainage plays a key role in deglaciation, and so their spatial characteristics, density, and distribution can help understand the patterns of glacial retreat (e.g., Sugden et al., 1991; Greenwood et al., 2007).

In spite of their importance, some outstanding questions remain: What are the typical lengthscales that characterize subglacial drainage systems (i.e., what is the drainage area, how many individual valleys form, how many tributaries do they have, etc.)? Can we reliably identify subglacial channels from rivers and other meltwater channels by using remote sensing techniques, including imagery and topographic data? And how do the characteristics of remarkably well preserved channels compare with channels elsewhere? To answer these questions, here we perform a detailed geomorphological study of exposed subglacial channels on Devon Island (Canadian Arctic Archipelago). This work represents the first field and high resolution remote sensing characterization of subglacial channels in the high Arctic, one of the areas with the best exposures of such features worldwide.

Well preserved exposed subglacial channels are rare. During glacial recession, meltwater released from the ice sheet accumulates at the ice marginal area and erodes the channel, with post-glacial sediment accumulation causing burial or partial burial (Le Heron et al., 2009). Vegetation overprint and fluvial incision makes the detailed study of channel geometry and morphology difficult (e.g., Walder and Hallet, 1979). Exceptions include areas with polar desert climate in the Antarctica Dry Valleys (e.g., Sugden et al., 1991) and the Canadian High Arctic (e.g., Dyke, 1993, 1999). The reduced rainfall conditions of these sites, recent ice retreat, and null or minimal vegetation cover are key for the preservation of these features. The morphology of subglacial channels at our field study area on Devon Island (Fig. 1), the second-largest of the Queen Elizabeth Islands in the Canadian Arctic Archipelago, is consequently well-preserved. The retreat of the Devon Island ice cap, in addition, offers a unique opportunity to compare recently exposed subglacial channels with systems incised during the Younger Dryas Innuitian de-glaciation.

The systems identified by Dyke (1999) as subglacial meltwater channels on Devon Island are ∼10-20m wide and $3-6$m deep, which is consistent with subglacial channels observed elsewhere, and one to two orders of magnitude smaller than tunnel valleys (e.g., Cofaigh, 1996; Kehew et al., 2012; Livingstone and Clark, 2016. Although subglacial channels have been described in detail in the field in northern Europe (e.g., Kleman, 1992; Clark et al., 2004; Piotrowski et al., 2006), the Antarctica Dry Valleys (e.g., Denton et al., 1984; Sugden et al., 1991), Canada (e.g., Kor et al., 1991; Beaney and Shaw, 2000; Shaw, 2002), and the United States (e.g., Walder and Hallet, 1979; Booth and Hallet, 1993), their rigorous distinction from other drainage systems from remote sensing imagery and topographic data is limited (Greenwood et al., 2007). Field

identification of subglacial channels consists on (1) identification from fluvial runoff (proglacial channels or river systems) and (2) distinction from other melwater features, primarily lateral meltwater channels (e.g., Beaney and Shaw, 2000; Greenwood et al., 2007; Syverson and Mickelson, 2009; Margold et al., 2013). A set of subglacial channel identification criteria is presented by Greenwood et al. (2007) and summarized in table 1. These guidelines are qualitative, and the characteristics listed here may or may not be all present in a set of subglacial channels (e.g., Sugden et al., 1991; Beaney and Shaw, 2000).

To categorize and characterize the features presented as subglacial meltwater channels in Dyke (1999), we conducted fieldwork on central Devon Island. In section 2 we provide a detailed description of our field data acquisition and processing. We acquired GPS borne channel longitudinal profiles along with stereo imagery and derived photogrammetry digital elevation models. We also used a Kinematic LiDAR Scanning (KLS) portable system, a novel method of measuring ultra-high resolution topography (<2cm/pixel Digital Elevation Models (DEMs)), which is described in more detail in Section 2.1.2. In section 3 we present a quantitative characterization of the observed channel networks, which we apply to distinguish subglacial channels from lateral meltwater channels and rivers in section 4. In section 5 we present a detailed qualitative description of subglacial channel morphology and the shape of subglacial drainage networks, which serves to further characterize and distinguish subglacial channel networks.

## 1.1 Field site: Devon Island

Devon Island was covered by an extensive Innuitian Ice Sheet that reached its maximum extent during the last glacial maximum (e.g., England, 1987; Dyke, 1999; England et al., 2006). Shortly after the Younger Dryas, around 10 radiocarbon years BP, the margin of this ice sheet began retreating towards the current coast line, and the final remnants in central Devon Island vanished around 8.8 radiocarbon years BP (Dyke, 1999), leaving a landscape of plateaus, fiords and deeply incised canyons. We refer to the work by Dyke (1999) and England et al. (2006) for a detailed discussion on the glacial history of the island during and since the Innuitian ice sheet. Since deglaciation, the landscape evolution is mainly the result of periglacial processes and erosion by ephemeral seasonal streams (e.g., McCann et al., 1972; Dyke, 1999). Fluvial incision represents only a small and highly localized contribution to the overall landscape evolution as a consequence of the island's polar desert climate conditions (e.g., French, 2013). Aerial photography obtained from the National Air Photo Library (National Resources Canada) of Devon Island reveals highly directional channel networks, which Dyke (1999) described as meltwater channels. These systems are incised into the otherwise flat plateaus that comprise the majority of the topography (regional slopes are 1 to 6°), and typically drain into deeply incised canyon systems that are believed to pre-date Innuitian glaciation (Dyke, 1999).

To categorize and study these meltwater channels in detail, we selected a study site that comprises an area of approximately 15 km$^2$ to the E-SE of the Haughton impact structure in central northern Devon Island (Fig. 1). This is a well-preserved 23 km diameter (Osinski and Spray, 2005) 23 Myr. old (Young et al., 2013) meteorite impact structure, which is a well established Mars analogue terrain, and has been the focus of numerous planetary analogue studies including crater morphology and erosion, periglacial landscape evolution on Earth and Mars, and evolution of ancient lake beds, among other activities (Lee and Osinski, 2005). The location provides access to both exposed subglacial channel and river networks, allowing for systematic comparisons of their geometry and longitudinal profiles. Geologically, the study area lies entirely within carbonate

strata of the Upper Ordovician Allen Bay Formation, specifically the Lower Member, which comprises a uniform succession of medium bedded to massive limestone with dolomitic labyrinthine mottling (e.g., Osinski and Spray, 2005; Thorsteinsson and Mayr, 1987), and is overlain by quaternary glacial till (e.g., Dyke, 1999; Osinski and Spray, 2005). Depositional landforms such as eskers, and glacial deposits including glacier moraines and striations are rare on the plateau surface of Devon Island (Roots et al., 1963) and only occur sporadically within the Haughton impact structure (Osinski and Spray, 2005).

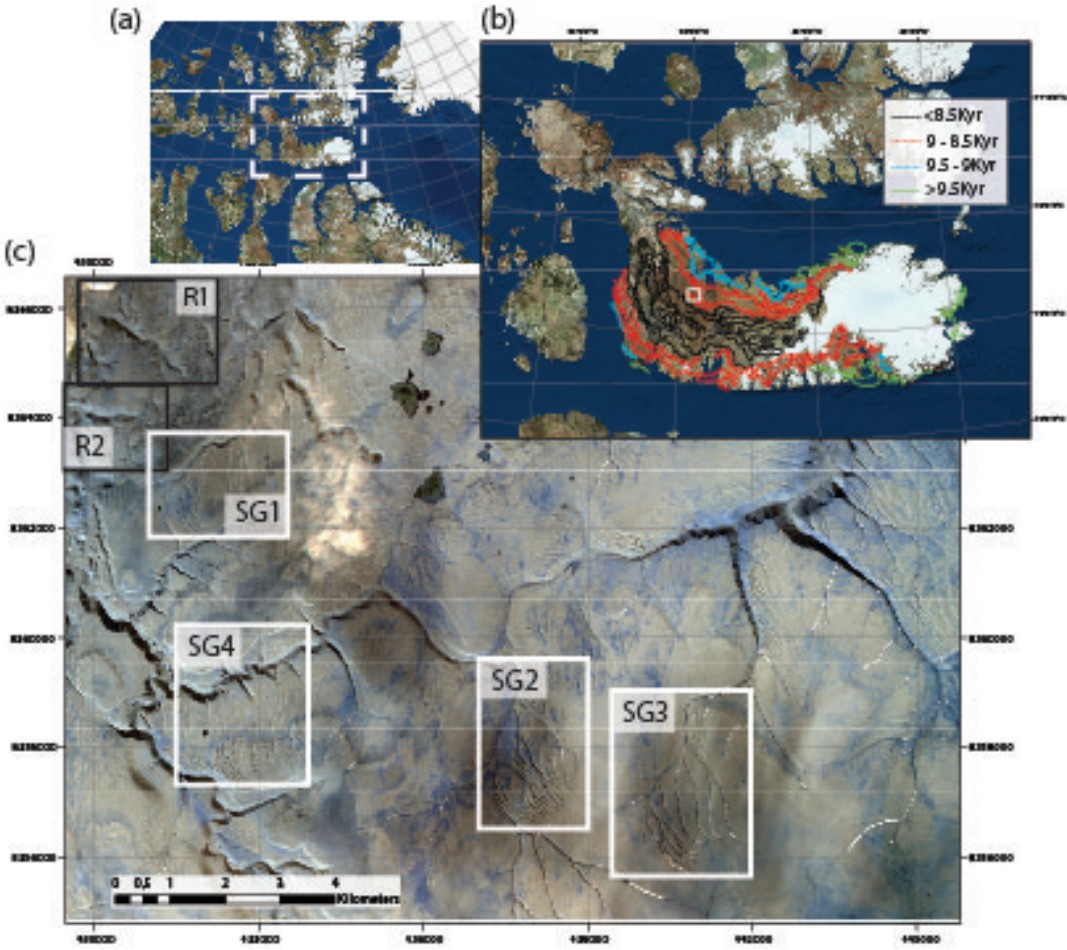

**Figure 1.** (a) satellite imagery of Devon Island within the Arctic Archipelago (white box). (b) satellite image of Devon Island, with a white box indicating the selected field site. The map also shows the Innuitian ice sheet terminus lines digitalized from Dyke (1999), with age reference in the legend (refer to radiocarbon years). (c) Field site (UTM zone 16), with boxes around each network investigated. White boxes are for sublglacial networks (SG1, SG2, SG3, and SG4), whereas black boxes indicate fluvial networks (R1 and R2)

## 2 Methodology

### 2.1 Preliminary remote sensing characterization

Figure 1(a) shows a Digital Globe satellite image of the Canadian Arctic including Devon Island and the rest of the Arctic Archipelago for context. Also shown is the Innuitian ice sheet termini as digitized from the work by Dyke (1999). Figure 1(b)

shows a high resolution satellite view of our selected field area. Our target locations consist on 4 distinct subglacial channel networks and 1 fluvial network. To identify the areas with potential subglacial incision, we looked for highly directional channel networks parallel to former ice flow lines in areas easily accessible from the Haughton structure, and also based on the locations where Dyke (1999) found evidence for meltwater channels. Using high resolution WorldView imagery of the site (resolution of 2 m/ pixel), CDEM (0.15 arc-sec DEM corresponding to 20 by 36 m/ pixel at a latitude of $75°$ N, obtained

from the Natural Resources Canada website (http://geogratis.gc.ca/site/eng/extraction), and the recently released Arctic DEM (release 5 at a resolution of 2 m/pixel available for free at https://www.pgc.umn.edu/guides/arcticdem/distribution/) we chose 6 channel networks, from which we inferred 4 to be subglacial and 2 to be fluvial.

From these 6 channel networks, we selected and visited 20 individual channels for detailed, in situ characterization (see Fig. 1). For this study, we selected only first order tributaries and characterized them from the origin until the first junction. In

most occasions, meltwater accumulates into streams and fluvial incision is apparent from field and remote sensing data in the profiles and cross sections of the channels once the network develops a stream order of 2 or more. In addition to in situ data, we acquired helicopter airborne imagery of sites located in central and eastern Devon Island, and identified subglacial channels as they are exposed by the current retreat of the ice cap at its terminus.

### 2.2 Longitudinal profile data

A distinctive characteristic of subglacial channels is the presence of vertical undulations in their profiles (e.g., Sissons, 1961; Sugden et al., 1991; Greenwood et al., 2007). To detect these features we obtained longitudinal profile data (i.e., elevation vs. distance data) of the 20 target channels using a GARMIN gpsmap 64s with a 1-3 m horizontal resolution, depending on polar satellite availability, and a vertical barometric resolution of 3-6 m. We acquired the data by walking or driving an All Terrain Vehicle (ATV) along each inferred river or subglacial channel from the head until the first junction, and averaging the multiple

profiles acquired in two to three runs.

To minimize the effects of the variable GPS resolution in processing the data, we grouped the channels into 5 groups that correspond to sites visited on the same day during the season. We also recorded the speed of the traverse during the collection of the longitudinal profiles, which varied between walking speed ($v_{walk} \sim$ 3-5km/h) and ATV speed ($v_{drive} \sim$ 10-15km/h). We then use this information to filter the GPS data corresponding to each field day (Fig. 3).

We processed the GPS raw data by high pass filtering the signal with an upper frequency of twice the Nyquist sampling size, which corresponds to 1 GPS point per 3 seconds in the hiking traverses and 1 point per second in the ATV traverses, and a lower frequency corresponding to the inverse of the time required to drive/ walk along the channel length, that is, $L/v_{drive}$

in ATV traverses and $L/v_{walk}$ in hiking traverses (where $L$ is the channel length). This filtering operation removes the data spikes related to avoiding obstacles on traverses including stream paths, large boulders, and snow patches.

## 2.3 Airborne imagery and photogrammetry

We complemented our in-situ channel characterizations with an extensive collection of high resolution aerial photography of over 50 subglacial channels throughout the island, from which we derived Digital Surface Models (DSM). DSM generation through stereo-photogrammetry processing of image data involves the reconstruction of a three-dimensional body employing measurements in two or more overlapping images, acquired from different positions. Accurate reconstructions require an overlap of more than 50% between each image on a basis of at least 10 images per location, common features identifiable in different images for reference, and detailed spatial coordinates for each site.

For this purpose, we acquired over 1000 helicopter airborne images to capture the topography of multiple inferred subglacial channel networks. To build this image database, we used a GPS-referenced CANON EOS 6D with an image resolution of 72ppp (5472 by 3648 pixels) (e.g., Smith et al., 2009). The built-in GPS has a horizontal spatial resolution of $\sim 10$m and a vertical resolution of $\sim 5$m. Although the camera GPS resolution also depends on polar satellite availability, resolution variations are minimal given the very small time lapse of image acquisition of all helicopter data.

To construct a Digital Surface Model (DSM), we obtained geo-referenced helicopter borne images for more than 50 channels including the centre-east of the island and the margin of the Devon Island ice cap. Significantly, this survey includes imagery of subglacial channels currently emerging under the active Devon Island ice cap margin (Fig. 2(c)), which enabled us to ground truth our identification scheme. Figure 2 includes examples of these images acquired at different points in the island.

We process the data in several steps. For each site, we first upload the images into AGISOFT software (e.g., Tonkin et al., 2014), together with the camera-generated EXchangeable Image Format (EXIF) files that include the geo-reference information. The software automatically aligns the imagery using the overlap existent between images. We improve the initial automated alignment with manual alignment of the images by selecting and matching common features (control points). Next, we produce a dense point cloud, a meshed surface model, and a surface model with ground texture. At this stage, we use the recently released Arctic DEM (available for free download at http://www.agic.umn.edu/arcticdem) to manually introduce markers in the model with known coordinates and elevation. This step improves the resolution of the final product by an order of magnitude. With this improved 3D model, we produce an orthophoto and the Digital Surface Model (DSM). In turn, at the final stages of processing we manually crop the DSM to remove noisy areas.

The DSM model reconstructions range in resolution between 0.4 m/pixel to 10 m/pixel depending on helicopter elevation and speed, number of images captured at each site and their overlap, the number of manually introduced control points, and other factors. All products are available upon request from the authors in point cloud (.LAS) and Geotiff (.tif) formats.

## 2.4 Kinematic LiDAR Scan acquisition

We used a novel kinematic backpack LiDAR scanning (KLS) system to capture the detail of subglacial channel topography from the ground. This study is the first time the KLS system has been deployed in the Arctic, and the first time it has been

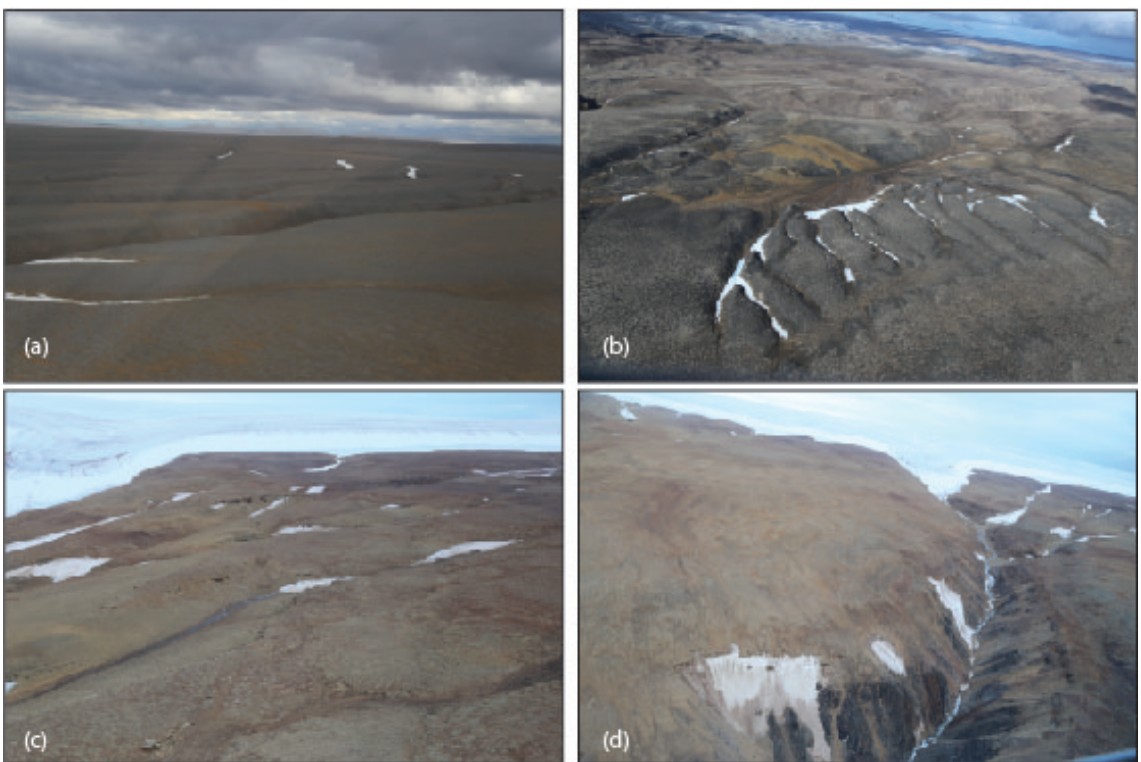

**Figure 2.** Aerial and field imagery of subglacial channels and rivers. 2(a) corresponds to helicopter imagery of a group of subglacial channels (89.13° W,75.28° N), channel widths approx. 35 m. 2(b) corresponds to a groups of subglacial channels located at 89.37° W, 75.18° N, network is approx. 300 m wide. 2(c) corresponds to a subglacial channel emerging underneath the Devon Island ice cap, notice the similar morphology to 2(a) and 2(b), each channel being approx. 30m wide. 2(d) shows the cross section of a deeply incised canyon emerging from under the ice cap. Canyon cross section measures 200 m approx.

applied to detailed subglacial channel morphometric measurements. The goal of the survey was to reproduce the surface topography at cm resolution, but also to make a proof of concept for the capabilities of kinematic LiDAR. The system consists of a LiDAR scanner, a GNSS/GPS positioning system, and an inertial measurement unit (IMU) which are mounted within a backpack frame, allowing the user to make ultra-high-resolution LiDAR point clouds ($>5{,}000\ \mathrm{points/m^2}$) of features traversed by the user. KLS enables the reproduction of surface topography at cm resolution (2 cm/pixel; Fig. 4), which is a higher level of accuracy compared to e.g. airborne LiDAR data, which would also be expensive to obtain in remote areas such as the high Arctic. The KLS dataset and derived surface models improve cross sectional and longitudinal profile analysis and comparison with river valleys, and are a ground-truth for hand-held GPS topographic data acquisition.

The LiDAR data was acquired using the AkhkaR3 kinematic backpack LiDAR developed at the Finnish Geospatial Research Institute, which is an updated version from those presented in Kukko et al. (2012) and Liang et al. (2015). The system is based on GNSS-IMU (Global Navigation Satellite System-Inertial Measurement Unit) positioning system, consisting on Novatel

SPAN: Flexpak6 receiver, UIMU-LCI inertia measurement unit and 702GG antenna, a 360 degrees of field of view cross-track profiling laser scanner (Riegl VUX-1HA) synchronized to the positioning and operated by a tablet computer (Panasonic Toughpad ZF-G1).

River and subglacial channel topography data were collected by traversing the centreline of the channel by ATV, with the operator carrying the LiDAR system on his/her back. Continuous scanning was done using 150 Hz profiling, and 500 kHz pulse repetition frequencies. With these settings the maximum range was about 200 m from the scanner (i.e. a 400m-wide channel could be completely scanned), and the along-track line spacing about 1 cm with angular resolution of 1.8 mrad.

For accurate trajectory computations we set up a GNSS base station (Trimble R10) at the Haughton river valley base (75° 22.42' N, 89° 31.89' W) constituting of about 5 km base line length to the target channels. The raw GNSS observables were recorded at 5 Hz frequency, as were those at the KLS mapping system. The altitude data for the KLS system were recorded at 200 Hz data rate, and the positions and altitude trajectory were computed in a post-processing step for the point cloud generation. For post-processing we used the base station position using PPP method and the tightly coupled KLS trajectory Waypoint Inertial Explorer 8.60 (NovAtel, Canada).

To produce an elevation model the raw point cloud data was further processed: the points resulting from multiple reflections were removed as well as points with weak return signal (intensity less than 800 in the scanner scale). Some remaining points resulting from the laser beam hitting the rear of the ATV during the capture were manually cleaned out of the data using CloudCompare software. Post-processed data was exported as .LAS files.

The final DEMs (see example in Fig. 4 (c) and (d)) from the georeferenced LiDAR point clouds were created using .LAS Dataset tools in ArcGIS (LAS Dataset to Raster: Bin, Avg, Simple). The effective pixel resolution of 2cm/pixel in the DEMs represent the average value of the point cloud within a 2 cm bin (typically 5-10 LiDAR points, depending on proximity to the scanner). Due to the very high spatial coverage of the LiDAR points, minimal interpolation was needed, except in areas of LiDAR shadow. These areas were interpolated using the simple interpolation method outlined in the ArcGIS help section.

## 3 Results: Quantitative characterization of river and subglacial channels

### 3.1 River and subglacial channels' longitudinal profiles

Reliably identifying subglacial channels on the basis of the criteria listed in Greenwood et al. (2007) and summarized in table 1, and in particular recognizing sections where the subglacial flow eroded against topographic gradients (e.g., Sissons, 1961; Sugden et al., 1991; Glasser et al., 1999) requires measuring longitudinal profiles in the field. Figure 3 shows the longitudinal profiles of different channels visited within the field area (see Fig. 1 and Table 2 for location reference). In this figure, SG Network 1 consists of a group of inferred subglacial channels. R Network 1 is an inferred river system with 2 investigated channels, and R Network 2 includes 4 investigated river channels. SG Network 2 - D group is a group of inferred subglacial channels, with 2 channels investigated, and so is SG Network 3 - P group. SG network 4 - CAMF group consists of inferred subglacial channels, of which we visited 5.

From these data we identify the channels that display undulating profiles, as well as quantify the maximum undulation in each profile which we define as the elevation difference between the local minima at the beginning of a section with positive topographic gradient $h_{min}$, and the local maxima that follows it $h_{max}$, to the total topographic loss $\Delta H$, that is: $\psi = (h_{max} - h_{min})/\Delta H$ (see Fig. 10 for a cartoon representation). We refer to this magnitude as the magnitude of undulation $\psi$ and we

use it herein to quantify differences between the fluvial and subglacial longitudinal profiles. Table 2 shows the magnitude of undulation $\psi$ of the different channel networks in Fig. 3, together with their detailed coordinates.

Channels in SG Network 1 (j201, j202, j204, j233, and j234) show magnitudes of undulation $\psi$ corresponding to 24%, 3%, 27%, $\psi = 3\%$, and $\psi = 0\%$ respectively of the total topographic loss, which is equivalent to 6.6 m, 1.5 m, 6.5 m, 1.6 m, and 0 m respectively for j201, j202, j204, j233, and j234. Channels j201 and j204 display the largest undulations that we analysed.

R Network 1 corresponds to inferred fluvial systems, which we investigated and analysed for comparison. In this network, no undulations are detected above the GPS confidence level, and in fact the profiles show a steady decrease of elevation with distance that is much more consistent with profiles of fluvial channels in the literature (e.g., Howard, 1994; Sklar and Dietrich, 1998; Whipple and Tucker, 1999; Whipple, 2004). R Network 2 shows two profiles corresponding to active rivers (j231 and j232), which show $\psi = 4\%$ and $\psi = 0\%$ respectively. Subglacial channels in SG Networks 2 and 3 were identified

on the basis of their similar morphology and orientation to subglacial channels in other networks, but do not present significant undulations. Within SG Networks 2 - D and 3 - P, channels j251 and j253 display undulations of $\psi = 7\%$, j252 has $\psi = 1\%$, and j254 displays $\psi = 3\%$. Finally, SG Network 4 - CAMF displays 5 channels with different levels of undulation, respectively $\psi = 0\%, \psi = 13\%, \psi = 4\%, \psi = 4\%$ and $\psi = 0\%$. Based on similar morphology and proximity to other channels with large $\psi$ within the same network, and recurrent $N - NW$ to $S - SE$ orientation, we conclude that all channels in SG Network 4 to

have originated in the subglacial regime.

Additionally, Fig. 3 provides a comparison of LiDAR (solid orange line) and GPS-acquired (dashed line) longitudinal profiles. This was a useful indicator of the reliability of the hand-held GPS profile dataset within the GPS resolution range. We discuss the LiDAR results in more detail below. We also performed an additional comparison of our data (LiDAR and GPS) with corresponding longitudinal profiles extracted from the Arctic DEM at 5 m/pixel resolution (Fig. 3 green lines). In most of

the profiles the agreement is excellent and well within the GPS precision margin. However, profiles j201, j211, j213, j214, j263 and j264 show significant deviations. Profiles j201, j202, j203, and j253 are also significantly noisier than the rest of Arctic DEM derived data. We attribute the discrepancy, in particular the difference in concavity in profiles j263 and j264 with our data, to the presence of snow covering the rivers and subglacial channels during the acquisition of the photogrammetry data used to derive the Arctic DEM. We did not observe in the field any of the spikes present in the Arctic DEM profiles j201,

j202, j253, and j254, and therefore we argue that they are DEM artefacts. However, this figure also proves that channel profile analyses based on Arctic DEM are reliable within the DEM limitations, and therefore subglacial channels can be identified and their undulations quantified using remote sensing high resolution topography.

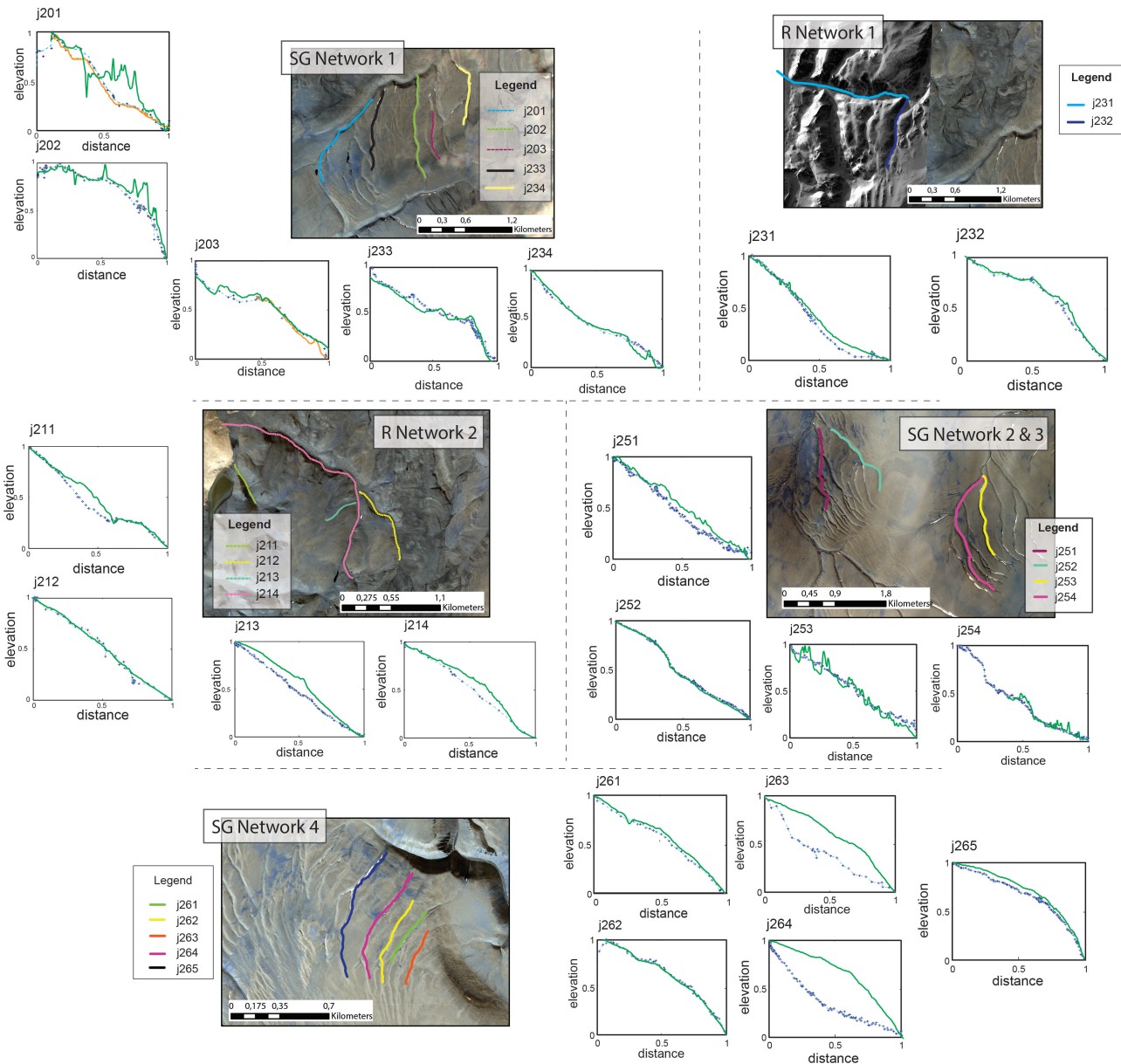

**Figure 3.** Longitudinal profiles of river and subglacial channels normalized to total topographic loss and length along the channel, with WorldView satellite imagery for each channel network. In the longitudinal profiles, blue crosses represent the raw GPS data for each channel, blue dashed lines are the data after filtering, and orange solid lines represent the LiDAR sections that overlap GPS data for comparison. The profiles obtianed with the Arctic DEM at 5 m resolution are shown in green color.

## 3.2 LiDAR observations

Kinematic LiDAR Scanning (KLS) was acquired in 5 subglacial channels and one river. The LiDAR dataset provides very high resolution topography data, which adds robustness to GPS-based undulation observations. Furthermore, KLS highlights a difference in cross sectional shape, scale, and downstream evolution that has not yet been considered as a distinctive characteristic of subglacial erosion, and is not appreciable from GPS profile data.

Figure 4 shows the results of using the kinematic backpack LiDAR approach to imaging the topography of a channel. The first panel 4(a) shows the point cloud files produced when investigating the channel cross section. The data is coloured by back scattered intensity at a laser wavelength of 1550 nm used in the KLS LiDAR system, resulting in darker values for wet snow and ice as seen in the image. Panel 4(b) shows the trajectory of the KLS user in blue lines overlapped to the point cloud product for a reference of coverage. Panel 4(c) shows the raster derived from the point cloud files for a river valley (corresponding to j231), and 4(d) shows the raster for the subglacial channels analysed in network 1, at a resolution of 9cm/pixel. Point spacing in 4(a) and 4(b) corresponds to 6 cm, with a total point count of 117,147,558 points for the river valley and for the subglacial channels. Raster resolution corresponds to 9 cm in the subglacial channels in network 1, and 10cm in the river valley corresponding to j231.

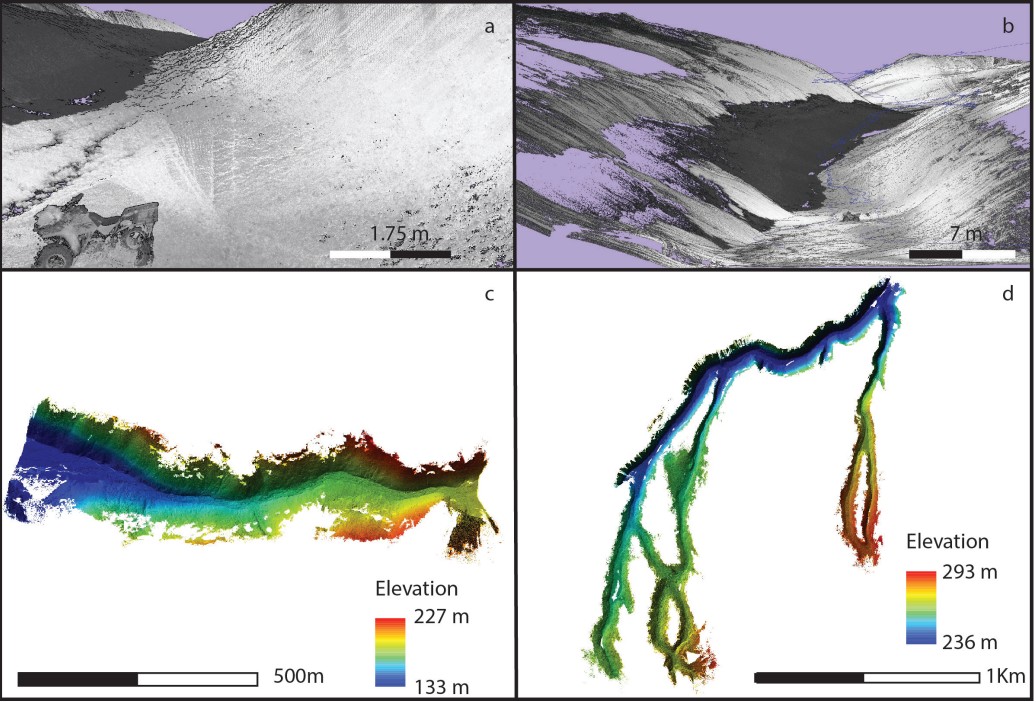

**Figure 4.** KLS LiDAR observations. Panels (a) and (b) show the color coded point cloud files (dark is low return), see the scale for spatial reference. Panel (b) shows the trajectory of the KLS user. Panels (c) and (d) show the raster produced using the point clouds.

Panels (c) and (d) offer a clear comparison of cross sectional scale, shape, and evolution in the case of river valleys (c) and subglacial channels (d). In (d), subglacial meltwater 1st order tributaries have widths of 5-7 m, and maintain a remarkably constant cross sectional scale as the channels evolve downstream. The cross-sectional shape is flat bottomed with steep-sided walls at the angle of repose ($\sim 15 - 20°$) as shown in panel (b) and described in more detail in next section. In comparison, the river valley cross section starts narrower but increases significantly towards the end of the channel, as shown in panel (c). Other features such as the absence of internal channels inside the subglacial channel flat bottoms are also evident in LiDAR observations.

## 3.3 Photogrammetry observations

DSM rasters produced with the technique described in section 2.1.2 allow for a detailed topographic study at higher resolution than the CDEM (30 m/pixel) or the Arctic DEM (2 m/pixel), but lower than the LiDAR observations. The advantage of this dataset over LiDAR and GPS observations is the mobility of the aircraft, which allowed for topographic data acquisition in different parts of the island. Stereo imagery and photogrammetry results include Digital Surface Maps (DSM), point clouds, and textured orthomosaics for an additional 10 subglacial channel networks. These datasets complement the LiDAR observations in different parts of the island at lower resolution, and are available at variable resolutions upon request as point clouds and Geotiff rasters. Figure 5 shows two DSM models and textured orthoimages of two different subglacial channel networks.

Surface models acquired through photogrammetry enable the differentiation of three regimes in a channel network (Fig. 5 (a)). In the first regime (zone (1)), subglacial tributaries originate as smooth depressions in the plateaus, merging into the topography and without clearly distinguishable heads. During the second zone (2), channels evolve into well developed systems $\sim 15$ m across and $\sim 4$ m deep in this network, keeping the width remarkably constant as they deepen downstream (see orthoimage below for better reference). Finally, in the last zone (3), tributaries merge into a deeper channel where fluvial incision by seasonal metlwater streams is apparent (notice the deepening in the DEM). In Fig. 5(b), DSM and orthoimage highlight the evolution of the tributary channels mostly by deepening as opposed to cross section widening (see particularly the scale of tributaries and main stem in the orthoimage). This high resolution local DEM highlights the size of the first order tributaries, which are $10 - 20$ m from the origin with no small scale channels or tributaries visible, their quasi-periodic spacing, and the smooth merging with local topography at the origin, which is an example of use of the topography data produced. Snow and ice accumulations were a common view in some channels, particularly closer to the Devon ice cap. This was an issue at processing the DSM and textured image, although in some channels the thickness of the snow pack could be estimated.

## 4 Identification of subglacial channel networks

## 4.1 Morphometric comparison of river and subglacial channels

Morphometric differences between rivers and subglacial meltwater channels are apparent even before analysing the longitudinal profiles in Fig. 3, only from the correlation to former ice margins and direction consistent with estimated ice flow lines of

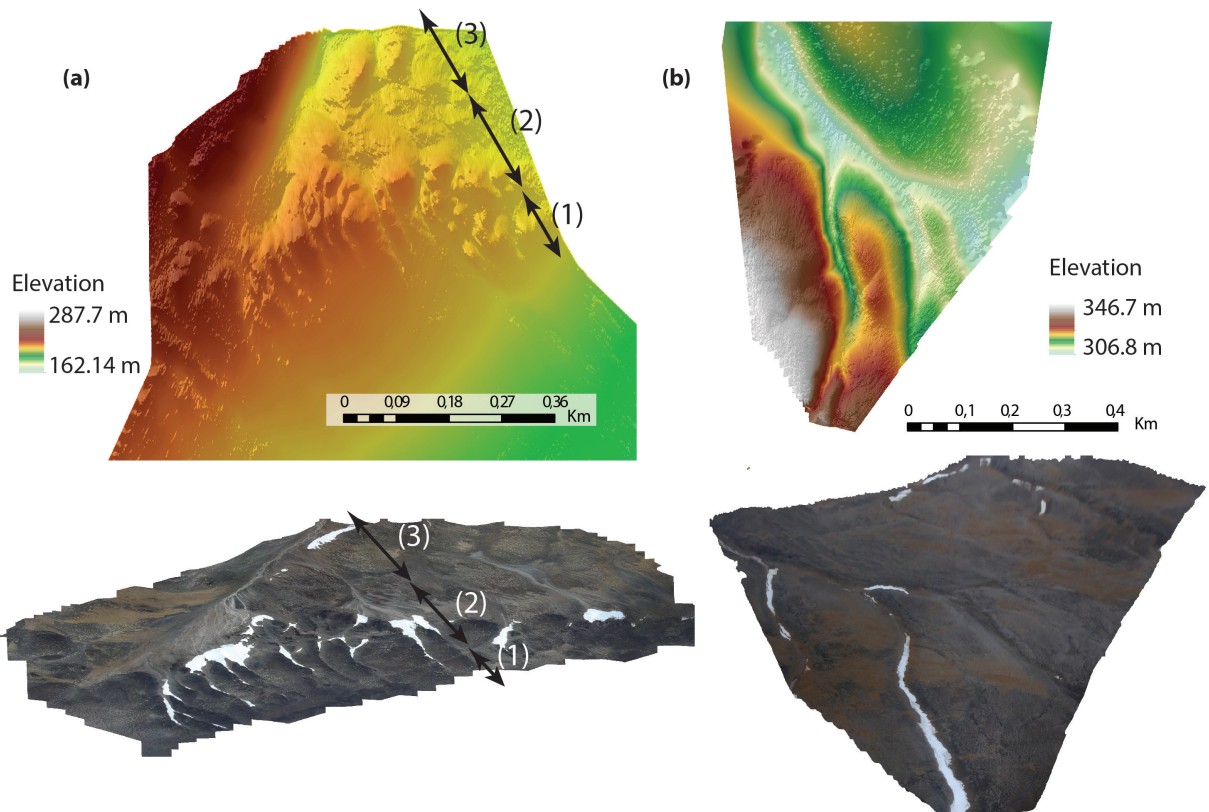

**Figure 5.** Stereo-photogrammetry derived from helicopter borne photography. The top panels (a) and (b) show the digital elevation model (DEM) at a resolution of 0.48 and 0.56 m/pixel respectively, with the colorbar indicating the elevation of the model surfaces. The images underlying the panels correspond to the textured orthoimages in both locations.

subglacial channels (Dyke, 1999). On remote sensing data of the field site and surrounding area, inferred subglacial meltwater channels appear in groups of $\sim 10$, parallel to each other consistently in the N-NW to S-SE direction. Moving to the east of the island, channel directions change on average from W to E, remaining oriented radially towards the current day Devon Island ice cap.

5     Characteristic inferred subglacial channel lengths are $\sim 1-2$ km throughout the distinct channel networks. The typical cross section is trapezoidal, with widths of $\sim 40-60$ m at the initiation stages ($\sim 150$ m downstream) that contain flat floors $\sim 20$ m wide, and depths of under 5m. Downstream ($> 1.3$ km), cross sections evolve to a better defined trapezoidal shape and deeper channels ($> 10$ m), preserving roughly the same width (Fig. 6).

     In comparison, the geometry of inferred river valleys on the island displays major differences. River widths vary continuously
10    downstream by one to two orders of magnitude from the origin ($\sim 5-10$ cm) until the first junction (60 m across, $\sim 5$ m deep)

∼ 150 m downstream from the headwater (i.e., Fig. 4(c) and (d); Fig. 6 initiation stages). Downstream, river valleys deepen up to ∼ 60 m and grow in width up to 400 m, forming deeply incised canyons with V-shaped cross sections (see Fig. 6, developed stage and Fig. 4). The evolution of bankfull channel width we observe is consistent with observations elsewhere (e.g., Parker, 1978a, b), and with the well-established hydraulic relationship for flow in river channels (e.g., Leopold and Maddock, 1953; Parker et al., 2007; Gleason, 2015), relating the channel bankfull width to the discharge:

$$W = K_w Q^b \tag{1}$$

Where for a gravel bed such as the ones considered, Parker et al. (2007) showed that $K_w = 4.63 g^{-7/10} D_{50}^{-5/2}$ and $b = 0.467$, with $D_{50}$ the medium value of the grain size distribution, and discharge that increases as tributaries merge into the main channel.

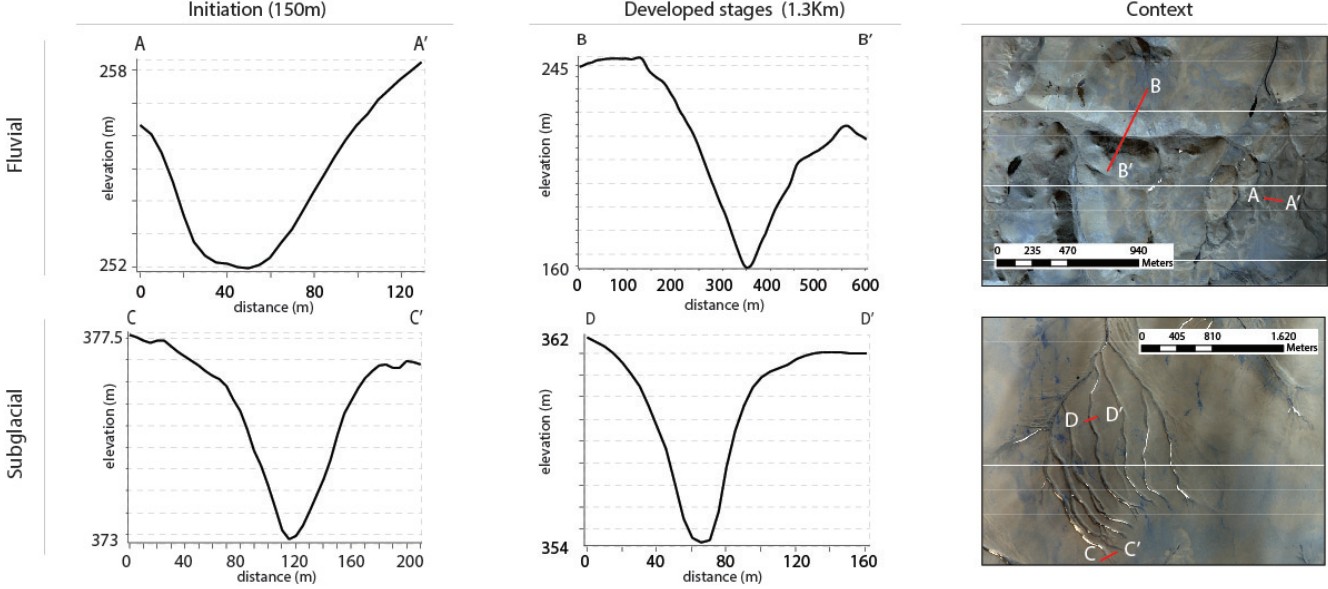

**Figure 6.** Cross sectional evolution of a fluvial (upper row) and subglacial (bottom row) channel, with satellite imagery for context on the right column. In the subglacial case, the initial width and the shape remain largely unchanged over length, whereas the river cross section grows monotonically both in width and depth with distance. Notice the differences in depth and length in the section scale bars.

The morphometrical characterization of the cross sectional differences between rivers and subglacial channels can be captured with a shape factor, $F = W_T/D$, defined as the ratio between channel top width and the depth (Leopold, 1970; Williams and Phillips, 2001) (see Fig. 10 for a reference cartoon). We measured channel top width following Grau Galofre and Jellinek (2017) as the distance between two points of maximum curvature along a cross sectional line for consistency, which corresponds with valley width for rivers and the width of the entire cross section in subglacial channels. We present shape factor results in table 2, column 6, where all measurements correspond to cross sections before tributary junctions. These results highlight the fundamental differences between fluvial and subglacial cross sections: whereas subglacial shape factors are in

the range $4.5 - 31$, with an average $<F_{SG}> \pm \sigma = 13.5 \pm 9$, fluvial shape factors are much smaller, in the range $2 - 4.6$ with an average $<F_{SG}> \pm \sigma = 3.4 \pm 1$. Furthermore, according with our observations in figure 6, we expect the variation of this shape factor to be considerable for subglacial channels as the top width remains constant and the channel deepens, and less important for river valleys, as both cross section and width increase downstream.

Another geometrical distinction between both erosional regimes is the width of first order tributaries (Grau Galofre and Jellinek, 2017). Even at the tip of the channel, subglacial channel widths are up to tens of metres (consistent with arguments in Weertman (1972)), as opposed to widths of first order river channels which are typically sub-meter in scale (Grau Galofre and Jellinek, 2017).

## 4.2   Comparison of lateral and subglacial meltwater channels

Along with the distinction from river systems, it is relevant to distinguish subglacial channels from channels formed by meltwater accumulated and released at the ice margins, i.e., lateral meltwater channels (i.e., Greenwood et al. (2007); Syverson and Mickelson (2009); Margold et al. (2013)), which have also been identified in the area (Dyke, 1999). We follow the criteria presented in Table 1, Greenwood et al. (2007), to this end. Focusing now only on the meltwater channel networks and ignoring rivers, the systems we investigate present a number of characteristics that exclude lateral meltwater drainage: (1) they do not

follow contour lines, but rather run parallel or slightly oblique to topographic gradients (Fig. 7) (e.g., Price, 1960; Greenwood et al., 2007); (2) their longitudinal profiles often contain stepped sections (e.g., Sugden et al., 1991; Greenwood et al., 2007) (Fig. 3), and may or may not display significant undulations (Fig. 3); (3) they display anastomosing patterns, with channel sections that split in two to join again further downstream (anabranching) (e.g., Sugden et al., 1991); and (4) potholes and shallow depressions are a common sight (e.g., Sugden et al., 1991; Greenwood et al., 2007). Figure 7 below shows a hillshade

map with contour lines representing each of the networks under study, both derived from Arctic DEM stripes at a resolution of 5 m/pixel. The boxes at the bottom right corner of each figure give information about the direction of the regional slope (red arrow) and the channel direction (black arrow).

    Comparing both directions, and taking into account that the channel networks we study are perpendicular to, and feed into, large canyons (see networks 1, 2, and 4) instead of forming terraces at their rims, we conclude that a lateral meltwater origin is

25 unlikely. We discuss more details regarding the morphology and characteristics of these networks in the next subsection, which also suggest the emplacement of these features in subglacial conditions.

## 5   Detailed morphology of subglacial channels in Devon Island

Based on the field and remote sensing observations presented above, we build a detailed description of the morphology of the 4 networks of subglacial meltwater channels we visited while in Devon, as specified in Fig. 1 and Fig. 7.

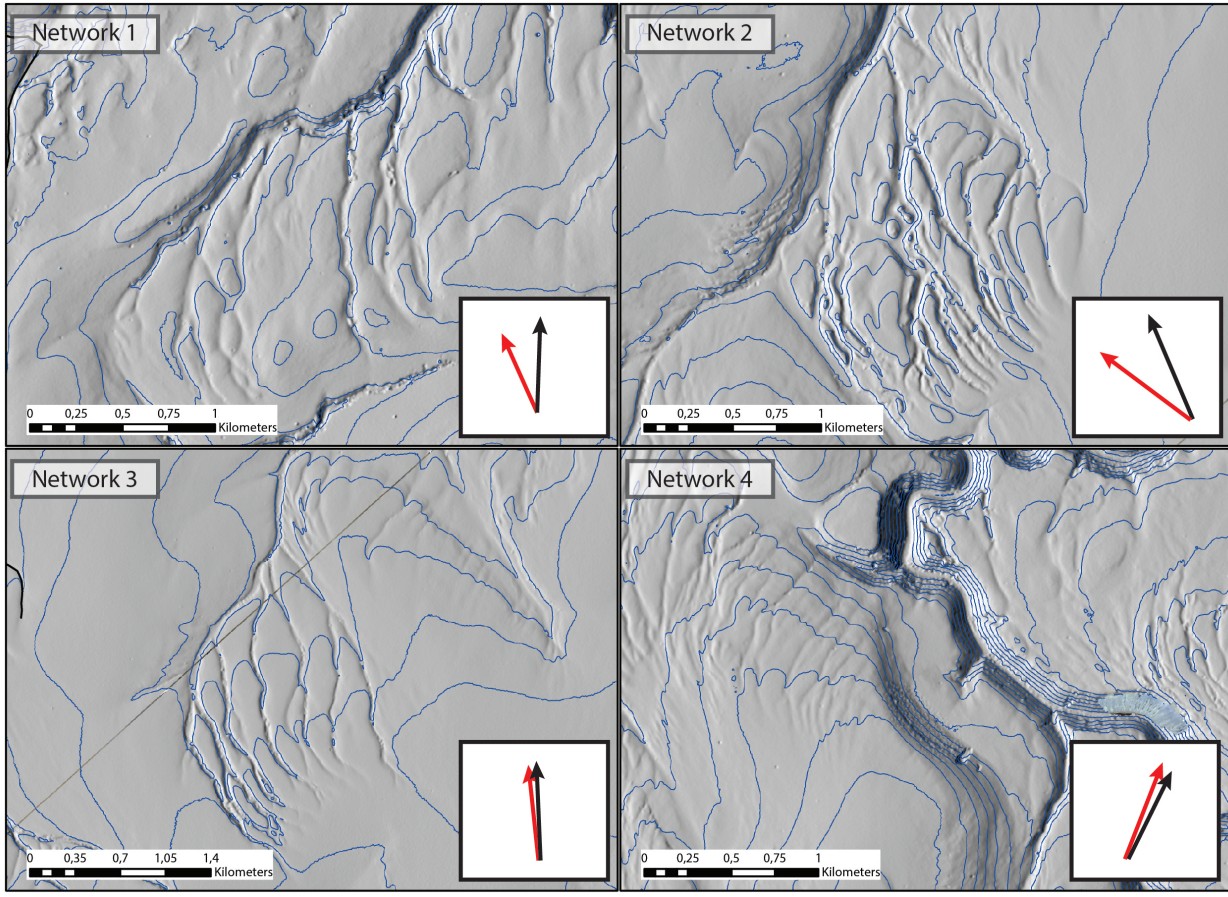

**Figure 7.** Hillshade and contour map of the 4 subglacial channel networks investigated. Contour lines are separated 15 m, and hillshade resolution is 2 m/pixel. In the bottom right corner, the black arrows indicate the overall direction of the channels in the networks, whereas red indicates the regional slope direction.

## 5.1 Network characteristics

The overall channel systems range from $2.5$ to $0.9$ km long and $1.6$ to $1$ km wide. They are all incised into dolomite bedrock and coarse gravel. Regional slopes in the plateaus where the networks are incised are very small (see table 3), and the number of tributaries varies from 5 in network 4 to 17 in network 2, consistent with networks elsewhere in the island. Table 3 contains a summary of morphological field observations of subglacial channels.

The general pattern of the 4 subglacial channel networks studied is dendritic (in that channels merge to produce larger channels) with a main channel that wraps around the exterior of the network and tributaries that flow parallel to it, merging

at acute angles (see networks 1 and 3 in Fig. 7), which gives the system of channels a finger-like appearance. In a few cases in networks 1, 2, and 3, tributaries bifurcate to give the network an anastomosing pattern. All subglacial channel networks observed terminate in deeply incised canyons that predate glaciation through hanging valleys or very steep chutes.

We observed at different sites on the island how the melting of snow accumulations within subglacial channels leads to the formation of meltwater ephemeral streams, which merge at the channel junctions into larger streams. This transition is often associated with a gradual change in cross section, from the shallow flat-bottomed form characteristic of subglacial channels to a deeply incised V-shape. Fig. 5 exemplifies this morphological transition from a network of subglacial channels to a single meltwater fed river.

## 5.2 Channel characteristics

### 5.2.1 Main channel

Main stems are $1.5 - 2$ Km in length and follow a NE-SW direction for about $1$ Km before bending around tributaries. The profiles of these channels are stepped, with steps consisting on 2-3 segments about $300 - 500$ m long separated by sections of steeper gradient (Fig. 3, see channels j201, j252, j254). In some occasions, there is a short section of reverse gradient following these steps. Main stem cross sections are generally trapezoidal, with flat bottomed floors and steep sided walls ( $\sim 20°$ degrees). Variations in width from the channel origin until the junction are small, accounting for no more than a few meters of change in any of the networks (c.f., Fig. 4 and fig. 5).

### 5.2.2 Tributaries

All tributaries in each network typically formed at the same elevation and incised the substrate parallel to, or oblique to, the topographic gradients to meet the main channel (Fig. 7, table 2 column 5). Typically, the subglacial channel networks we observe consist of 10 to 17 tributaries oriented in the NE-SW direction. Within the same network tributary depth can vary between $< 1 - 10$ m. This differential incision is particularly evident in networks 1 and 2, although this property is noticeable in all networks within the resolution of this study (Fig. 8).

In general, tributaries display the same trapezoidal flat bottomed, steep sided shapes that are characteristic of the main channels, although cross sectional asymmetries appear here with more frequency than in the main stems. In particular, tributaries in network 3 are incised more deeply in the eastern side than the western side, allowing for shallow depressions to form along the steeper side (Fig. 8, left panel). Developed tributaries (at a distance of $\sim 1$km from their origin) are around $\sim 10$ m in depth, with a flat floor $\sim 10$ m across and steep sided walls up into the plateaus, as shown in the examples of fig. 6.

Longitudinal profiles of tributaries are complex and vary across the different networks (Fig. 3). Tributaries in networks 1 and 4 grade into the main channel continuously, whereas in networks 2 and 3 some confluences present hanging valleys followed by shallow potholes in the junctions between smaller and larger tributaries (see an example in fig. 9 (c)). Profile curvature is variable even within the same network: channels j202, j232, j261, j262, j265 display concave profiles, whereas channels j231,

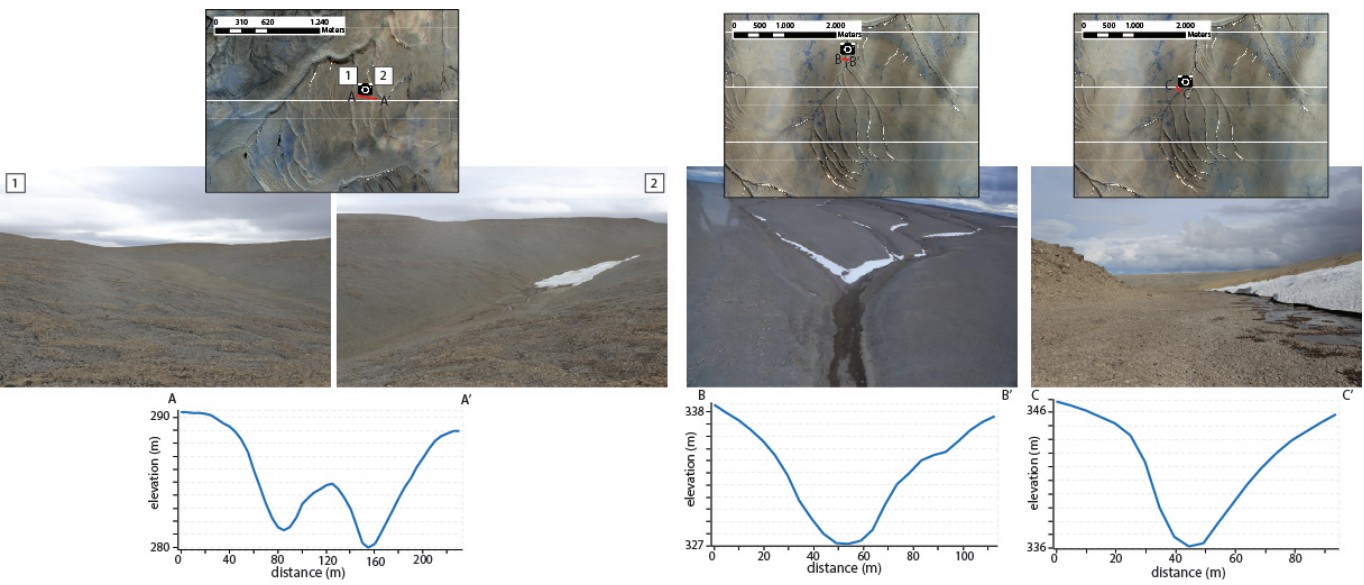

**Figure 8.** Cross section field imagery and profiles. Upper row shows a satellite imagery context on the location where the image and the cross section are obtained, together with a scale reference. Middle row shows images of four cross sections, obtained by this expedition on July 24th and 25th, 2017. The middle panel corresponds to a main channel whereas the other three images correspond to tributaries. Cross section profiles below show elevation (m) vs. distance (m) obtained from the Arctic DEM at 2 m/pixel.

j234, j251, j263, and j264 display convex profiles. Shallow potholes ($\sim$ 1m deep) filled with water or snow are a common view across all networks, as detailed later.

## 5.3 Other characteristics

### 5.3.1 Anastomosing patterns

5 Although the shape of the networks is mostly dendritic (channels merge to produce larger channels), anabranching (bifurcation followed by re-junction downstream) patterns occur frequently at the beginning of the networks, typically before 1 km. Examples of this anabranching behavior are shown in networks 1, 3, and 4, where channels split to rejoin anywhere between $5 - 250$ m downstream. Figure 7 shows a hillshade map of the subglacial networks, where the anastomosing patterns are easily identified. Also of interest here, the left panel in fig. 8 shows a high resolution cross section taken across an anabranching section

10 in network 1. The section clearly shows how one of the channels (in this case the eastern channel) is more deeply incised than the western one, which may suggest a time-transgressive emplacement of the system (e.g., Beaney and Shaw, 2000; Brennand, 1994).

### 5.3.2 Potholes

Potholes appear frequently in the subglacial channel networks explored in the field (Fig. 9), and they are also evident from the photogrammetry and LiDAR DEMs we produced, falling at resolution edge of the Arctic DEM. They are shallow depressions (50cm to 2 m deep) typically filled by water that grade into the channel floors, and that vary in dimensions between $\sim$ 5-50m long by $\sim$ 2-30 m across, with a particular example in network 1 where dimensions are up to $\sim$ 125 m and $\sim$ 40 m across (Fig. 9 panel (b)). We observed them to occur (1) at the junction of two tributaries, (2) in the middle of a tributary channel associated with a channel widening, or (3) at the channel headwater area. Typically, the larger sizes appear in case (1), whereas the smaller depressions occur in (2). Figure 9 shows field images of potholes of several sizes, fitting into type (1) (panel c and b), class (2)(panel d), class (3) (panel a). The location of these features is indicated in the aerial images at the side with a camera icon.

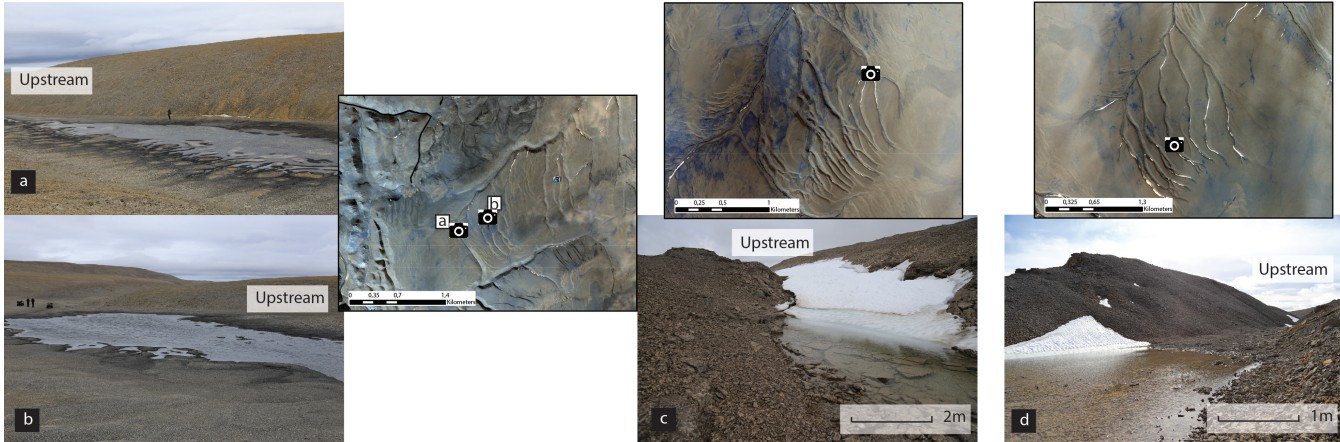

**Figure 9.** Field images of the shallow depressions and potholes observed. Satellite imagery provide context for the photographies through the camera icons. In photos (a) and (b), notice the human figures for scale. Photos (c) and (d) contain a scale bar for reference. Image (c) is an example of an overhanging valley (here covered in snow) followed by a pothole.

## 6 Discussion

### 6.1 Undulations, obliquity, shape factor, and the remote sensing characterization of subglacial channels

In section 3.1 we introduce the magnitude of undulation $\psi$ and discuss its role in identifying subglacial channels. This is a useful metric, but it requires the acquisition of very high resolution topographic data. At lower resolution, in addition, differences among channel direction and local topographic gradients can also be indicative of subglacial erosion, as long as ice erosion rate by sliding is lower than meltwater erosion rate (e.g., Weertman, 1972; Shreve, 1972; Paterson, 1994). Observations of channels incised oblique to topographic gradients are common in the literature (e.g., Sissons, 1961; Walder and Hallet, 1979;

Sugden et al., 1991; Livingstone et al., 2017). Quantifying and measuring these deviations in a set of subglacial channels to stablish a quantitative base for channel categorization has, however, not been done.

Considering the confined flow of pressurized water in a subglacial channel at the base of an ice sheet to follow the x direction, with y perpendicular to ice flow and z perpendicular to the ground surface, so that $z_b$ and $z_i$ are the bed and ice surface elevations respectively. At steady-state, water flow at the base of the ice is driven by the water pressure potential gradient $\nabla\phi$:

$$\nabla\phi = -\rho_i g \nabla z_i - \Delta\rho g \nabla z_b + \nabla N. \tag{2}$$

Here $\rho_i$ is the ice density, $\Delta\rho = \rho_w - \rho_i$ is the density difference between water and ice, $g$ is the gravity, and $N = p_i - p_w$ is the effective pressure, where $p_i = \rho_i g(z_i - z_b)$ is the local hydrostatic pressure related to ice thickness and $p_w$ is the water pressure. The topography of Devon Island's plateaus is mostly flat, which implies that the controls in effective pressure gradient arise mostly from variations in ice surface slopes, $\nabla z_i$, and not from surface topographic gradients $\nabla z_b$. This picture is true generally if ice surface slope is more important than bed topography, such that $\rho_i g \nabla z_i / \Delta\rho g \nabla z_b >> 1$. Although ice surface slope is correlated with topography at a regional scale it can depart from topography at the scale of individual channels (Fig. 10), driving both channelized and distributed meltwater accordingly. This explains the slight deviations we observe between subglacial channel direction and local topographic angles in our field site, recorded in table 2.

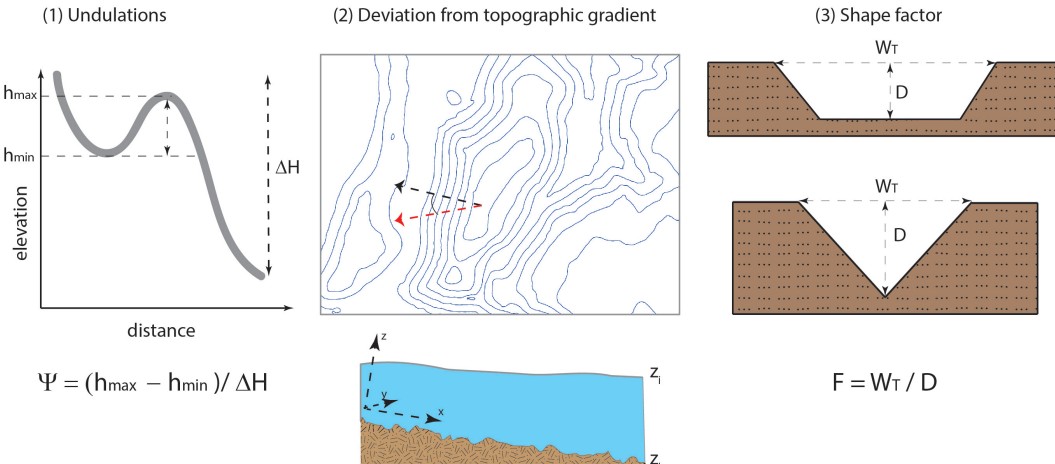

**Figure 10.** Cartoon representing the definitions of the three remote sensing based metrics proposed in this study. (1) Shows our definition for longitudinal profile undulations $\Psi$, where the grey line represents the longitudinal profile of a channel (elevation vs. distance). (2) Represents the deviation between the direction of a set of channel networks (red arrow) and the topographic gradient (black arrow), together with the axis notation and the ice and topographic surfaces $z_i$ and $z_b$ in equation 2. (3) Shows the definition of shape factor with two cartoons representing a trapezoidal and a V-shaped cross section, where top width $W_T$ and depth $D$ are represented (adapted from Williams and Phillips (2001)).

However, where ice topography is nearly flat or bed slopes are important, $\rho_i g \nabla z_i / \Delta\rho g \nabla z_b << 1$, bed topography dominates incision and drives meltwater flow. Under these conditions, undulations or departures from topographic gradients cannot occur. In this case, neither metric will identify channels as subglacial, and their characterization will depend on other observations,

such as cross sectional characteristics or morphology. Morphological criteria include the presence of anabranching patterns, consistent direction with former ice flow lines, and correlation with other subglacial features (i.e., eskers, moraines, outwash fans) (Greenwood et al., 2007; Kehew et al., 2012, e.g.,).

## 6.2 Identification of subglacial channels from remote sensing data

We distinguish subglacial channels in our field area on the basis of four properties, which are measurable at the Arctic DEM resolution: (1) consistent N-NW to S-SE direction, radial to the paleo-ice margins (Dyke, 1999, Fig. 8) near our field area, changing to W to E near the ice cap margin; (2) Topographic undulations in the longitudinal profiles (Fig. 3 and Table 2), and channel incision with orientations not parallel to the local topographic gradient (Fig. 7); (3) Cross-section size and shape, i.e., shallow trapezoidal for subglacial channels and deeply incised V shape for river valleys(Fig. 6 and Fig. 8, quantified in table 2 column 6 in the shape factor), which evolve in rigorously distinguishable ways (Fig. 6; and (4) large 1st order channel widths on the order of $\sim 10$ m (Fig. 4). Not all channels we identify as subglacial from their morphology and direction have undulations in their longitudinal profiles. However, none of the river profiles show any detectable undulations. We conclude that the magnitude of the undulation index $\psi > 0$ unequivocally distinguishes subglacial erosion, but $\psi = 0$ does not necessarily preclude it. Similarly, deviations from local topographic gradients for subglacial channels are much larger ($> 5°$) than for rivers.

We add three remote sensing indicators of subglacial erosion to the criteria in Greenwood et al. (2007). The first indicator is the large cross-section widths at the origin of order 1 channels (i.e., Fig. 4) which are orders of magnitude larger than in river systems. The second criterion is the minimal variation in width downstream, from the beginning of the channel until the first junction, as visible from Fig. 4, comparison of panels (c) and (d), and Fig. 6. The third is the remarkable difference between shape factors (top width to depth ratios) between river valleys and subglacial channels.

## 7 Conclusions

In this study we describe a population of subglacial channels (N-channels) exposed on Devon Island, Canadian Arctic Archipelago. In particular, we implement, and discuss the use of, remote sensing techniques to distinguish between systems of rivers, lateral meltwater channels, and subglacial channels, which serves as a complement to existing field based methods. We provide detailed field descriptions of 20 individual channels, including their longitudinal profile characteristics, cross sectional geometry, channel directionality and drainage network morphology, to then revisit and expand the identification methods listed in Greenwood et al. (2007). Our field observations include GPS mapping of subglacial and fluvial incised channels longitudinal profiles, photogrammetry, kinematic LiDAR data (KLS), and aerial imagery, allowing for both a qualitative and quantitative description.

Subglacial channels appear in clusters of 10-17 parallel individual systems that roughly follow the topographic gradients (Fig. 7). Tributaries and main stems are wide, $\sim 40 - 60$ m at the initiation stages, with defined trapezoidal shapes that are relatively shallow ($5 - 10$ m) and preserve roughly the same width downstream. Other patterns characteristic of subglacial

drainage, such as anastomosting networks, potholes, and hanging valleys at the junctions, are also common in the channels investigated.

We find that a quantitative measure of undulation $\psi$, defined as the topographic loss (local minima to local maxima of the undulation) at an upstream section to the total topographic loss, reliably distinguishes fluvial and subglacial longitudinal

profile (Fig. 4), although the lack of undulations does not rule out subglacial erosion. We also argue that the departure in channel direction from local topographic gradients also reflects subglacial erosion (table 2), as well as a large top width-to-depth ratio (shape factor F, compiled in table 2.) We then discuss the limitations of both these metrics in identifying subglacial channels. If both metrics fail, other morphological observations such as channel direction, anastomosing networks, cross sectional scale and downstream evolution, serve as a categorization guide.

With our observations, we present the first high resolution study of the subglacial drainage channels of the high Arctic, as well as revisit well-established field identification guidelines (e.g., Greenwood et al., 2007). We conclude with the following target characteristics of interest: (1) undulations in the longitudinal profile and changes in channel direction with respect to local topographic gradients, (2) consistent channel direction following former ice flow lines, close to the ice margin, and with the possible presence of other subglacial features in the area, (3) order 1 channel widths on the 5-10 m scale with minimal

variation downstream, with wide and trapezoidal cross sections, and (4) presence of anabranching sections.

*Data availability.* All datasets described in this paper are available upon request from the lead author (contact agraugal@eos.ubc.ca). This includes photogrammetry generated digital elevation models (.tif format), LiDAR data (.tif and .LAS formats), airborne geo-referenced imagery (.jpg format), and GPS elevation tracks (.txt format).

*Author contributions.* Anna Grau Galofre and A. Mark Jellinek designed the field campaign and collected hand-held GPS and photogram-

metry data. G. Osinski is the principal investigator for this campaign, provided logistical and technical support, and aided in collecting hand-held GPS data. A.Kukko and M.Zanetti collaborated equally in collecting and processing LiDAR data. The body of the manuscript was written by A. Grau Galofre, with additions from all collaborators. The authors declare that they have no conflict of interest.

*Acknowledgements.* This work has benefited by stimulating discussions with E. Godin, G. Flowers, I. Hewitt, C. Schoof, and G. Clarke. This work was supported by an NSERC Discovery grant to A.M. Jellinek, an NSERC DIscovery Grant Northern Supplement to G. R. Osinski, and

a NSERC CREATE fellowship to A. Grau Galofre. Arctic DEMs were created from DigitalGlobe, Inc., imagery and funded under National Science Foundation awards 1043681, 1559691, and 1542736. LiDAR studies were made possible by financial funding from the Academy of Finland for "Multi-spectral personal laser scanning for automated environment characterization (300066)" and "Centre of Excellence in Laser Scanning Research (CoE-LaSR) (272195)". We thank the Polar Continental Shelf Project (PCSP) for their logistical support without which this work would not have been possible.

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

**Table 1.** Diagnostic criteria for the identification of subglacial channels

| Distinctive characteristics | references |
| --- | --- |
| Undulations in the longitudinal profiles | Sissons (1961); Greenwood et al. (2007); Kehew et al. (2012), this study |
| Direction oblique to topographic gradient | Sissons (1961); Sugden et al. (1991); Greenwood et al. (2007), this study |
| Presence of other subglacial landforms | Greenwood et al. (2007); Kehew et al. (2012) |
| Cavity systems and potholes | Sugden et al. (1991), this study |
| Stepped confluences | Sugden et al. (1991), this study |
| Abrupt beginning and end | Sissons (1961); Glasser et al. (1999) |
| Absence of alluvial fans | Sissons (1961), this study |

| Other characteristics | references |
| --- | --- |
| Presence of abandoned loops | Clapperton (1968), this study |
| High sinuosity | Clapperton (1968) |
| Bifurcating and anastomosing patterns | Clapperton (1968); Sugden et al. (1991); Greenwood et al. (2007), this study |
| Variety of size within the same system | Sissons (1960) |
| Presence of steep chutes | Sissons (1961), this study |

**Table 2.** Morphometric characteristics

| Individual channel | latitude | longitude | $\psi$ | deviation from topography | $F$ |
|---|---|---|---|---|---|
| SG Network 1 | | | | 27° | |
| j201 | 75° 20' 0" | 89° 25' 5" | 0.24 | | 4.5 |
| j202 | 75° 20' 3" | 89° 23' 13" | 0.03 | | 8 |
| j204 | 75° 20' 11" | 89° 22' 49" | 0.27 | | 7 |
| j233 | 75° 20' 14" | 89° 22' 19" | 0.03 | | 10 |
| j234 | 75° 20' 02" | 89° 24' 44" | 0 | | 10 |
| R Network 1 | | | | - | |
| j211 | 75° 21' 35" | 89° 27' 40" | 0 | | 2 |
| j212 | 75° 21' 28" | 89° 28' 42" | 0 | | 3 |
| j213 | 75° 21' 12" | 89° 25' 22" | 0 | | 4 |
| j214 | 75° 21' 25" | 89° 27' 10" | 0 | | 4 |
| j215 | 75° 21' 04" | 89° 26' 24" | 0.01 | | 3 |
| R Network 2 | | | | - | |
| j231 | 75° 20' 56" | 89° 27' 5" | 0.04 | | 4.5 |
| j232 | 75° 20' 25" | 89° 27' 39" | 0 | | 3 |
| SG Networks 2 - D | | | | 30° | |
| j251 | 75° 17' 27" | 89° 11' 41" | 0.07 | | 5.5 |
| j252 | 75° 17' 39" | 89° 9' 42" | 0.01 | | 30.5 |
| SG Networks 3 - P | | | | 4° | |
| j253 | 75° 16' 43" | 89° 4' 55" | 0.07 | | 6 |
| j254 | 75° 17' 06" | 89° 5' 17" | 0.03 | | 5.5 |
| SG Network 4 - CAMF | | | | 6° | |
| j261 | 75° 17' 28" | 89° 27' 41" | 0 | | 17 |
| j262 | 75° 17' 23" | 89° 27' 48" | 0.13 | | 17.5 |
| j263 | 75° 17' 23" | 89° 27' 26" | 0.04 | | 31 |
| j264 | 75° 17' 24" | 89° 28' 01" | 0.04 | | 18 |
| j265 | 75° 17' 24" | 89° 28' 22" | 0 | | 17 |

**Table 3.** Summary of morphological characteristics

| Characteristic | SG network 1 | SG Network 2 - D | SG Network 3 - P | SG Network 4- CAMF |
|---|---|---|---|---|
| tributary $n°$ | 10 | 17 | 10 | 5 |
| regional slope (%) | 1.8 | 2.0 | 1.3 | 5.5 |
| plungepools? | yes | yes | yes | yes |
| anabranching sections? | yes | yes | yes | no |
| hanging valleys? | no | yes | yes | no |
| network length (km) | 1.5 | 2.1 | 2.5 | 0.9 |
| network width (km) | 1.6 | 1.3 | 1.5 | 1 |
| stepped profiles? | yes, j201 | yes, j252 | no | no |
| network shape | dendritic | dendritic | dendritic | parallel |
| presence of other subglacial bedforms | no | no | no | no |