# Peer review of "Subglacial drainage patterns of Devon Island, Canada: Detailed comparison of rivers and subglacial meltwater channels"

_The Cryosphere, 2017_

## Referee Comment (RC1) · S.J. Livingstone (Referee) · 22 Dec 2017

**Review | Subglacial drainage patterns of Devon Island, Canada: detailed comparison of river and tunnel valleys. Galofre et al. (2017)**

General Comments

This is an interesting paper that presents original research on a series of bedrock cut meltwater channels on Devon Island, Canadian Arctic Archipelago. A number of techniques including GPS, mobile LiDAR data and stereo imagery derived DSM are applied to investigate the difference between subglacial meltwater channels and rivers at high spatial resolution. There is a lack of high resolution morphological analysis of bedrock carved meltwater channels and this is therefore welcome work. The paper itself is generally well written and structured. However, I do have a number of general comments that I would like to see addressed before this paper is published, and some more specific comments below this.

1. *Use of the term tunnel valley*: The term tunnel valley is traditional used to refer to much larger features of the order of several kilometres wide and tens of kilometres long, that may be cut into sediment or bedrock. The features described here seem to be an order of magnitude smaller and I therefore suggest sticking to the term subglacial meltwater channel or N-channel throughout.

2. *Missing literature:* A large body of work on subglacial meltwater channels, including how to identify them in the geological record (e.g. Greenwood et al., 2007 and references therein) and their morphological properties and spatial distribution (e.g. Brennand & Shaw, 1994; Kristensen et al., 2007; Livingstone & Clark, 2016 to name but a few) seem to have been missed, with a lot of emphasis instead given to the Kehew et al. (2012) paper. In the discussion at least I was expecting the authors to refer back to previous work to put into context how these features are similar or different. Indeed, in the discussion, the text on the hydraulic potential equation is presented as original work, whilst it is actually well known (see Shreve, 1972), and their 'new' metric for tunnel valley identification on channel directionality is not really new (e.g. see Greenwood et al., 2007). The authors may also want to look at and compare their work to some of the recent modelling work that has tried to incorporate fluvial erosion into numerical ice models to investigate the formation of N-channels.

3. *Morphology of the subglacial meltwater features*: I believe this paper really undersells what is a fantastically high resolution study of the morphology of bedrock carved channels. I am not aware of such detailed work in such well preserved landforms and yet the results seem rather hidden away after the comparison of the different techniques. I would like to see more made (and example figures shown) of the channel morphologies, including further discussion of the headwalls, anabranching pattern, spacing, cross-sectional profiles and association with other bedforms, while a summary statistics table would also really help the reader. As currently written, it is the use of the different techniques which really comes out from this, not the morphology of the features. To broaden this work out it would have been nice to see how their dimensions compare with other studies of similar sized features (and then also the larger tunnel valleys) and to discuss what this means in terms of their formation (e.g. slow and steady vs catastrophic drainage).

Specific Comments

P1L1: Tunnel valleys can also be cut into sediment.

P1L6: should be "extent"

P2L28: I think there needs to be some recognition of the different scales here. N-channels are typically associated with much smaller channels cut specifically into bedrock. Tunnel valleys/channels may also be cut into sediment and are much large. In terms of the effect on ice dynamics – most of the work is associated with the evolution to channelized drainage and these channels are again envisaged to be an order of magnitude smaller than tunnel valleys/channels.

P2L10: "from" instead of "only with".

P2L26: e.g. in wrong position in brackets beginning "Denton…"

P2L27: See also for a comprehensive mapping along a large portion of the southern sector of the Laurentide Ice Sheet: *Livingstone, S.J. and Clark, C.D., 2016. Morphological properties of tunnel valleys of the southern sector of the Laurentide Ice Sheet and implications for their formation. Earth Surface Dynamics, 4(3), p.567.* Indeed, there is a large body of work in this area, and also in the North Sea: see older papers in review by Kehew et al. (2012) – for completeness it would be good to reference some of the key work.

P2L34: This is conjecture – where is the evidence for temporal variability and large inputs?

P3L2: Although there has been a large body of work on the morphology of tunnel valleys (e.g. Livingstone & Clark, 2016).

P3L15: "ice sheet began retreating towards the current…"

P3L17: Capitalise Ice Sheet.

P5L11: This is not obvious from Fig. 1.

P5L25: What about figures 2 and 3?

P9L1-5: It would be useful for the reader if you included a schematic, perhaps on one of the profiles in Fig. 3 as it is not clear to me.

P9L19: "to have originated in a subglacial regime."

P9L8: "tributaries have widths of…"

P9L8-10: I also found that apparent anabranching of the channels an interesting feature worth observing. In particular, can you tell from the DEM whether the channels were formed synchronously (same depth of channel bottom), or time-transgressively (which might manifest as different depths of anabranching channels).

P13L1-2: More details are needed here. How does the cross-sectional shape and depth change between recognised tunnel valleys and river channels? This is a key distinction that has been glossed over here.

P13L3: This is not clear to me as only a small portion of the image corresponds to tributaries that you pick out as having similar widths. Is there a better example?

P13L10: Braiding is the wrong term here I believe as this would refer to temporary islands as part of a dynamic sedimentary system. Anabranching is a more appropriate term

P13L8-11: Again, this seems very short on details. You state that you can pick out the key characteristics of tunnel valley networks but then seem to restrict this to a few choice observations.

P13L13: "criteria exposed before" is an odd phrase. Re-write.

P13L19: delete "targeted"

P13L21: "and approximately constant downstream from the origin until…"

P13L17: "tunnel valley widths are up to tens…"

Do these channels merge into the surrounding topography at their origin or do they have a clear amphitheatre-headed canyons? (e.g. see Lamb et al., 20016, 2014). This might give you some clues as to their origin.

P13L34: "Examples of this pattern are shown in…"

P14L1: "tree-like network typical of…"

P14L3: "tunnel valleys also have very few."

P14L8-21: This is nicely summarised, but not new. The hydraulic potential gradient has been widely used to infer channel direction and we know that the ice surface slope can drive water over topographic undulations.

P15L4: And critically, their morphology and association with other subglacial features like eskers, moraines, outwash fans.

P15L5: What is the example and how does that help? The text below does not mention other subglacial bedforms.

P15L10-12: I am not convinced this is a new metric for tunnel valley identification (e.g. see Greenwood et al., 2007; Livingstone & Clark, 2016 – section 3.1).

P15L11-12: I do not understand this final sentence. If the ice is cold based surely large meltwater channels are unlikely to form?

P16L5: How can you be so precise in stating the timing of these features?

Figures:

Figure 1: Can you distinguish, maybe with different colour arrows, between the river valleys and tunnel valleys. This would help the reader.

Figure 2: In the caption you refer to distinct groups but these are not clear from the figure. It would be useful to include these headings so the reader can easily distinguish. It is not that obvious from the profiles why some have been termed tunnel valleys and some rivers. For instance, most of group 4 and 5

tunnel valleys are relatively smooth with little in the way of reverse bed slops, and are therefore comparable to groups 2 and 3. What allowed you to distinguish these as tunnel valleys rather than river channels?

Figure 4: Missing a colour legend for panels (c) and (d).

Figure 5: What do the arrows refer to? More details on what is actually picked up in these images would be helpful to the reader.

References:

Beaud, F., Flowers, G.E. and Venditti, J.G., 2016. Efficacy of bedrock erosion by subglacial water flow. Earth Surface Dynamics, 4(1), pp.125-145.

Brennand, T.A. and Shaw, J., 1994. Tunnel channels and associated landforms, south-central Ontario: their implications for ice-sheet hydrology. Canadian Journal of Earth Sciences, 31(3), pp.505-522.

Greenwood, S.L., Clark, C.D. and Hughes, A.L., 2007. Formalising an inversion methodology for reconstructing ice-sheet retreat patterns from meltwater channels: application to the British Ice Sheet. Journal of Quaternary Science, 22(6), pp.637-645.

Kristensen, T.B., Huuse, M., Piotrowski, J.A. and Clausen, O.R., 2007. A morphometric analysis of tunnel valleys in the eastern North Sea based on 3D seismic data. Journal of Quaternary Science, 22(8), pp.801-815.

Lamb, M.P., Howard, A.D., Johnson, J., Whipple, K.X., Dietrich, W.E. and Perron, J.T., 2006. Can springs cut canyons into rock?. Journal of Geophysical Research: Planets, 111(E7).

Lamb, M.P., Mackey, B.H. and Farley, K.A., 2014. Amphitheater-headed canyons formed by megaflooding at Malad Gorge, Idaho. Proceedings of the National Academy of Sciences, 111(1), pp.57-62.

Livingstone, S.J. and Clark, C.D., 2016. Morphological properties of tunnel valleys of the southern sector of the Laurentide Ice Sheet and implications for their formation. Earth Surface Dynamics, 4(3), p.567.

Shreve, R.L., 1972. Movement of water in glaciers. Journal of Glaciology, 11(62), pp.205-214.

Stephen Livingstone

---

## Referee Comment (RC2) · M. Margold (Referee) · 3 Jan 2018

The study of Grau Galofre et al. is an attempt to provide quantitative data for distinguishing the traces of former subglacial conduits in the glacial geomorphological record. There have been few such studies in the past and it is thus a commendable effort. The study is technically well-executed but it largely ignores relevant available literature, to its own detriment. A major issue, already raised by Stephen Livingstone (Reviewer 1) is an erroneous use of the term 'tunnel valley' while the landforms in question would be described by most as meltwater channels. Treating the studied landforms as tunnel valleys, and referring almost solely to literature on tunnel valleys (which are features of much larger size as S. Livingstone points out), the authors put all the weight on distinguishing between their 'tunnel valleys' – but meaning subglacial meltwater channels – and fluvial channels. However this is a simplistic approach. When attempting to better reconstruct and understand the characteristics of the former drainage systems of glaciers and ice sheets, it is as important as distinguishing between subglacial meltwater and fluvial channels (i.e. traces of former subglacial drainage vs record of more recent fluvial drainage) to distinguish subglacial meltwater channels from submarginal and purely lateral meltwater channels (i.e. traces of subglacial drainage vs traces of supraglacial, englacial and ice-marginal drainage). Indeed, many of the analysed channels that display low undulation in their longitudinal profile and occur in series cut in the slope might have formed as submarginal or true lateral channels. There is an ample amount of literature on these: both older, largely descriptive studies (e.g., Mannerfelt, 1945; Mannerfelt, 1949; Sissons, 1958; Clapperton 1968) and newer attempts to classify and discuss glacial meltwater features (Greenwood et al., 2007, 2016). Of particular interest might be a study of Syverson and Mickelson (2009), with observations of modern-day meltwater channel formation, and a study by Art Dyke (1993) who used meltwater channels to reconstruct the character of glaciation and the pattern of deglaciation in the broader region of this submitted manuscript.

Submitting this review as a second reviewer and having read the review by Stephen Livingstone, I concur with all his comments.

ISSUES TO ADDRESS: The motivation for the study and its setting within the context of existing knowledge in the field is not articulated enough. The authors might attempt to spell out more clearly what the study brings that older studies were lacking – this is a point where to refer to the existing literature on glacial meltwater channels.

The methods section is lengthy and at places self-serving. Why is there a need to reproduce the surface topography at cm resolution? The authors might attempt to better align the methods used with the stated objectives.

The distinction between 'tunnel valley erosional regime' and 'river valley erosional regime' is vague. Ideally, the authors might qualify the main characteristics of sub-glacial and fluvial drainage (based on literature) and look for the characteristic features in their data. The manuscript is overly relying on Kehew et al. (2012).

Portions of the text that refer to the figures read very much like figure captions.

Broader, v-shaped cross profile of the river valleys vs. narrower, flat-bottom, steep-walled cross profile of the meltwater channels – could something be inferred about the discharge and the length of formation/operation of the feature(s)? While this goes beyond the scope of the manuscript, a few references could be provided where this topic might be followed.

MINOR COMMENTS:

P1 L8 'Kinematic mobile LiDAR'. I am not an expert on this instrumentation but from checking briefly online, either one or the other adjective is usually used. Pairing the two adjectives seems to make little sense to me since they mean largely the same, just one having a Greek root and the other a Latin one.

P2 L23 Younger Dryas

P2 L33 Criteria is plural, write criterion where it is a singular.

P2 L34 Warm-based is a more common term than wet-based

P3 L14-17 Ages in Dyke (1999) are in radiocarbon years, however, the notation 'ka BP' is now commonly used for calendar years. Either state that it is C-14 years or calibrate.

P3 L20 remove the full stop before the reference

P3 L34 'deposition landforms' is a more common term

P5 L12-13 'Downstream of. . .' I don't understand what do you mean with this sentence.

P8 L24-25 Check the wording, 'represent' appears two times

P8 L32-33 'requires quantitative field longitudinal profile observations' reword to 'requires measuring longitudinal profiles in the field'.

P9 L24 agreement

P13 L3 replace 'in panel (b)' with 'in Fig. 5b'

P13 L4 replace 'packs' with 'accumulations'

P13 L13 'criteria exposed before' – exposed does not work here very well, search for a more fitting verb (stated, listed).

P13 L31 shallow

P13 L33-34 This is something that should be discussed further with references to older literature.

P14 L8-10 'We argue here that differences among channel direction and local topographic gradients are also indicative of subglacial erosion in areas where the ice erosion rate by sliding is lower than the meltwater erosion rate (Weertman, 1972; Paterson, 1994).' These can be submarginal meltwater channels that record the ice surface slope direction but do not necessarily bear any evidence with regard to ice erosion rate.

P15 L1-4 Here the fact that you have been ignoring all the types of meltwater channels other than subglacial really becomes problematic because you might be dealing with lateral or submarginal channels in this case.

15 L10-12 Again, there is a possibility that these might be submarginal or lateral meltwater channels. It might well be that most or all the channels that you classify as subglacial indeed are subglacial and not submarginal or lateral. But you need to be provide argumentation for this.

FIGURES: The figures are generally well-crafted.

Fig. 3 Add group numbers or panel letters so that the groups can be easily identified.

[Figure]

The figure would be more informative if one could see the topographic settings of the pictured groups of channels. Could the photographs possibly be draped on a DEM-derived hillshade?

REFERENCES:

Clapperton, C.M., 1968. Channels formed by the superimposition of glacial meltwater streams, with special reference to the East Cheviot Hills, North-East England. Geografiska Annaler. Series A. Physical Geography, pp. 207-220.

Dyke, A.S., 1993. Landscapes of cold-centred Late Wisconsinan ice caps, Arctic Canada. Progress in Physical Geography, 17(2), pp. 223-247.

Greenwood, S.L., Clark, C.D. and Hughes, A.L., 2007. Formalising an inversion methodology for reconstructing ice‐sheet retreat patterns from meltwater channels: application to the British Ice Sheet. Journal of Quaternary Science, 22(6), pp. 637-645.

Greenwood, S.L., Clason, C.C., Helanow, C. and Margold, M., 2016. Theoretical, contemporary observational and palaeo-perspectives on ice sheet hydrology: processes and products. Earth-Science Reviews, 155, pp.1-27.

Mannerfelt, C.M.S., 1945. Några Glacialmorfologiska Formelement: Och Deras Vittnesbörd Om Inlandsisens Avsmält-Ningsmekanik I Svensk Och Norsk Fjällterräng. Geografiska annaler, 27(1-2), pp. 3-5.

Mannerfelt, C.M.S., 1949. Marginal drainage channels as indicators of the gradients of Quaternary ice caps. Geografiska Annaler, pp.194-199.

Sissons, J.B., 1958. Sub‐glacial stream erosion in Southern Northumberland. The Scottish Geographical Magazine, 74(3), pp.163-174.

Syverson, K.M. and Mickelson, D.M., 2009. Origin and significance of lateral meltwater channels formed along a temperate glacier margin, Glacier Bay, Alaska. Boreas, 38(1), pp.132-145.
Martin Margold

---

## Short Comment (SC1) · 13 Jan 2018

I would like to thank Stephen Livingstone for this very detailed and constructive revision, which we believe will greatly enrich the manuscript. My coauthors and I agree on all the general revisions you proposed, and after considering the changes and literature sources you suggested, our plan is to modify the manuscript in the following way:

(1) Change the term 'tunnel valleys' to 'subglacial meltwater channels' throughout the manuscript

(2) Addition of literature relevant to the study

a. We will dedicate a subsection to the characterization of the meltwater channels as subglacial or lateral following the study by Greenwood 2007.

b.In the discussion section, we will also compare these features with meltwater subglacial channels identified in other parts of the globe, as suggested, including tunnel valleys. In particular, we will consider the length scales (longitude, cross section, and depth), presence and size of potholes, and the network geometry.

(3) Results subsection with a more detailed, qualitative description of the subglacial meltwater channels.

a. For this description, we will follow the structure used in Sugden 1991 and consider first the network scale, then the dominant channel, and then the tributaries. Here we will discuss spacing, although a detailed consideration of this particular observation will be the focus of another study.

b. As suggested, we will also consider in more detail the anabranching structure of the networks. We will make further observations from LiDAR and DEM data of the depth variation between the anabranching channels and include the time-dependence discussion suggested. We will consider adding a figure with cross sectional profiles along the anabranching section.

c. We will add to the paper the description of overdeepenings and potholes (with an image containing examples), and add details regarding the presence or absence of hanging walls and chutes in the channel junctions.

d. We will also include two figures with panels showing (1) images of the cross section at the site of channel initiation and downstream (i.e., >10 channel widths), and (2) cross sectional profiles of rivers and subglacial meltwater channels. This should address the lack of details regarding headwall geometry, cross sectional shape and evolution.

(4) Addition of a summary table, which will include the different channel networks visited (columns) and the presence/ absence of features characteristic of subglacial meltwater channels (rows, i.e., undulations, stepped profile, concave profiles, potholes/overdeepenings, etc.)

(5) Assess other minor specific comments.
* * *

---

## Short Comment (SC2) · 15 Jan 2018

We would like to thank Martin Margold for his thorough and constructive review. We believe that his suggestions, in particular considering submarginal and purely lateral meltwater channels in our channel characterization, will greatly enrich this manuscript.

It is likely that a number of channels in Devon are in fact lateral meltwater channels, so we believe that Dr. Margold raises a good point in advising the consideration of these drainage systems in our study. From the literature suggested in his comment (in particular Kleman et al. 1992, Greenwood et al. 2007, and Syverson and Mickelson 2009), it appears that a good test to distinguish between subglacial and lateral meltwater channels would be to delineate the channels we visited over a hillshade map with contour lines, where the reader can compare the overall direction of the channels under study with the topographic gradient. We will produce this map, which will motivate the characterization of subglacial vs. lateral meltwater channels.

The overall direction of the channels visited in the field roughly follows the topographic gradients, and we do not see examples of systems of channels subparallel to contour lines. Other morphological and geometrical aspects, such as the length scales (1-1.5Km long, several meters wide), anabranching patterns, presence of potholes, and perpendicular direction to the inferred retreating ice sheet margins (see fig.1 in the manuscript and Dyke 1999), are also suggestive of subglacial drainage (Sugden 1991, Greenwood 2007).

However, we do see examples of lateral meltwater channels from aerial imagery. Along deeply incised canyons, there are examples of channels incised on the side walls parallel to the canyon floor, in occasions forming series of nested channels. The morphology of these channels is consistent with a similar formation process to those investigated in Syverson and Mickelson 2009. There is a particular example in figure 2 panel (d) that follows the contour line along a valley, which is more in agreement with a lateral meltwater channel than a subglacial channel. We will remove this panel from the figure for consistency, but we will also consider adding an additional panel comparing subglacial and lateral meltwater channels to illustrate the discussion between these landforms Dr. Margold suggested.

In addition to making the distinction between lateral and subglacial channels, we will re-estate the motivation for this study to make it more specific to our results, and we will dedicate a subsection describing the details of the channel morphology (see response to Dr. Livingstone), both for subglacial and fluvial channels. There is an overall lack of remote sensing characterization of subglacial landforms, which this manuscript addresses at high resolution. In particular, the kinematic mobile (referring to its portability) LiDAR methodology introduced is new (see Kukko et al. 2012) and gives an insight into

the interior of the channels that DEMs and photogrammetry products cannot achieve, which is particularly relevant at detecting the presence/absence of inner channels or even potholes (Sugden 1991). These are, we believe, relevant additions to the body of data regarding subglacial landforms that motivate this study.

The last general point raised by Dr. Margold would be to include a discussion regarding the possible mechanism of formation, the length of operation, and the discharge accommodated by the channels identified. Although these aspects are indeed beyond the scope of the manuscript, and they will in fact be the direct focus of another study that is now in preparation, we can add a short paragraph in the discussion listing possible mechanisms and relevant references.

References

Dyke, A., 1999. Last glacial maximum and deglaciation of Devon Island, Arctic Canada: support for an Innuitian Ice Sheet. Quaternary Science Reviews, 18(3), pp. 393-420.

Greenwood, S.L., Clark, C.D. and Hughes, A.L., 2007. Formalising an inversion methodology for reconstructing ice-sheet retreat patterns from meltwater channels: application to the British Ice Sheet. Journal of Quaternary Science, 22(6), pp.637-645.

Kleman, J., 1992. The palimpsest glacial landscape in northwestern Sweden. Late Weichselian deglaciation landforms and traces of older west-centered ice sheets. Geografiska Annaler. Series A. Physical Geography, pp. 305-325.

Kukko, A., Kaartinen, H., Hyyppä, J., & Chen, Y., 2012. Multiplatform mobile laser scanning: Usability and performance. Sensors, 12(9), pp. 11712-11733.

Sugden, D. E., Denton, G. H., & Marchant, D. R., 1991. Subglacial meltwater channel systems and ice sheet overriding, Asgard Range, Antarctica. Geografiska Annaler. Series A. Physical Geography, pp. 109-121.

Syverson, K.M. and Mickelson, D.M., 2009. Origin and significance of lateral meltwater channels formed along a temperate glacier margin, Glacier Bay, Alaska. Boreas, 38(1),

pp.132-145.

---

## Editor Comment (EC1) · C. R. Stokes (Editor) · 22 Jan 2018

I would like to thank both referees for their constructive comments on this manuscript. The authors have provided a clear and thoughtful response and I would certainly be happy to consider a revised version of their manuscript.

Chris Stokes
* * *

---

## Author Comment (AC1) · 13 Feb 2018

We would like to thank the editor for considering this manuscript, and the two reviewers (Stephen Livingstone and Martin Margold) for the very detailed and thorough reviews, as well as constructive criticism. The comments of the reviewers made us particularly aware of the lack of detailed geomorphological characterization of the subglacial channels described, as well as the missing literature relevant to the topic, which hindered the interpretation and purpose of the paper. In response to the comments (see the supplement for our detailed response and relevant references), we added a geomorphological description of the subglacial channels, together with a discussion of lateral

meltwater drainage vs. subglacial drainage in the channels in consideration. The most significant changes in order of their appearance in the paper are:

1.- Following S. Livingstone's suggestion regarding the use of the term " tunnel valleys". We agree that the drainage systems described in this manuscript are considerably smaller and therefore should be referred to as subglacial channels. We also added a table (table 1) that summarizes the main characteristics of subglacial channels and puts our study in the context of the existing literature.

2.- Regarding our lack of consideration of lateral meltwater channels, we added a short subsection and a figure (subsection 4.2 and figure 7) where we discuss the relationship between the direction of the studied channel networks and the topographic gradients and regional slopes. We also quantified the angle between channel direction and regional slope in table 2.

3.- We added a new section, additional figures, and a table (section 5: Detailed morphology of subglacial channels in Devon Island, figures 8 and 9, table 3) where we present detailed field observations, including additional field photographs, and provide a much more elaborate description of subglacial channel morphology. Table 3 now includes a summary of observations for each subglacial network observed in the field as suggested by S. Livingstone.

We list the main comments of each reviewer and our corresponding response in the Supplementary PDF. We would like to thank the reviewers again for their thorough comments, which were helpful to improve this manuscript.

Sincerely,

Anna Grau Galofre, A. Mark Jellinek, Gordon R. Osinski, Michael Zanetti, and Antero Kukko.

Please also note the supplement to this comment:

https://www.the-cryosphere-discuss.net/tc-2017-236/tc-2017-236-AC1-supplement.pdf

**Supplement:**

**Response to the Reviewer's Comments**

Anna Grau Galofre      A. Mark Jellinek      Gordon R. Osinski
Michael Zanetti      Antero Kukko

February 13, 2018

We would like to thank the editor for considering the revised version of this manuscript, and the two reviewers (Stephen Livingstone and Martin Margold) for the very detailed and thorough reviews, as well as constructive criticism. The comments of the reviewers made us particularly aware of the lack of detailed geomorphological characterization of the subglacial channels described, as well as the missing literature relevant to the topic, which hindered the interpretation and purpose of the paper. In response to the comments, we added a detailed geomorphological description of the subglacial channels, together with a discussion of lateral meltwater drainage vs. subglacial drainage in the channels in consideration. The most significant changes in order of their appearance in the paper are:

1.- Following S. Livingstone's suggestion regarding the use of the term " tunnel valleys". We agree that the drainage systems described in this manuscript are considerably smaller and therefore should be referred to as subglacial channels. Consequently, we substituted " tunnel valleys" by " subglacial channels" throughout the manuscript, removed most of the references to [7] and other tunnel valley studies, and built the discussion around literature relevant to subglacial channels [e.g., 6, 8, 13, 14]. We also added a table (table 1) that summarizes the main characteristics of subglacial channels and puts our study in the context of the existing literature.

2.- Regarding our lack of consideration of lateral meltwater channels, we added a short subsection and a figure (subsection 4.2 and figure 7) where we discuss the relationship between the direction of the studied channel networks and the topographic gradients and regional slopes. Their direction perpendicular to canyon rims, roughly following topographic gradients, and radial to the ice sheet margins agrees with subglacial incision mechanisms. We also quantified the angle between channel direction and regional slope in table 2, and removed panel (d) in figure 2, as we acknowledge the possibility that this particular example is indeed a lateral meltwater channel.

3.- We added a new section, additional figures, and a table (section 5: Detailed morphology of subglacial channels in Devon Island, figures 8 and 9, table 3) where we present detailed field observations, including additional field photographies, and provide a much more elaborate description of subglacial channel morphology. Table 3 now includes a summary of observations for each subglacial network observed in the field as suggested by S. Livingstone.

We list the main comments of each reviewer and our corresponding response below. We would like to thank the reviewers again for their thorough comments, which were helpful to improve this manuscript.

Sincerely,

Anna Grau Galofre A. Mark Jellinek Gordon R. Osinski Michael Zanetti Antero Kukko.

**S. Livingstone** (in order of appearance, reviewer's comments in bold for easy referencing)

**Major comments:**

**1.- Use of the term tunnel valley : The term tunnel valley is traditional used to refer to much larger features of the order of several kilometers wide and tens of kilometers long, that may be cut into sediment or bedrock. The features described here seem to be an order of magnitude smaller and I therefore suggest sticking to the term subglacial meltwater channel or N-channel throughout.**

We addressed Stephen Livignstone's comment by substituting the term "tunnel valley" with the term "subglacial channel" throughout the manuscript. We agree with his correction regarding the length scales of the features here described (see also major comments)

**2.1.- Missing literature: A large body of work on subglacial meltwater channels, including how to identify them in the geological record (e.g. Greenwood et al., 2007 and references therein) and their morphological properties and spatial distribution (e.g. Brennand and Shaw, 1994; Kristensen et al., 2007; Livingstone and Clark, 2016 to name but a few) seem to have been missed, with a lot of emphasis instead given to the Kehew et al. (2012) paper. In the discussion at least I was expecting the authors to refer back to previous work to put into context how these features are similar or different.**

We addressed this issue throughout the paper. We added a whole section of subglacial channel morphological properties (see the list of major changes above) where we relate the field observations of channel characteristics with the work by [1, 3, 6, 8, 14, 16]. Acknowledging the first major comment, we also removed the emphasis on Kehew's work and support instead our categorization of subglacial channels on Greenwood's work (in particular table 1) and references therein.

**2.2.-Indeed, in the discussion, the text on the hydraulic potential equation is presented as original work, whilst it is actually well known (see Shreve, 1972), and their new metric for tunnel valley identification on channel directionality is not really new (e.g. see Greenwood et al., 2007).**

We corrected this part, which was indeed a badly worded section. We clarified this mistake by giving emphasis to the theoretical work done supporting the oblique direction of subglacial channels [e.g., 10, 12, 17] and also the observational evidence [e.g., 6, 13, 14, 16]. We highlight, however, the little attention that quantifying this metric has received, which allows for its use on morphometric channel comparisons. We added the numerical measure of this metric in table 2.

**2.3.-The authors may also want to look at and compare their work to some of the recent modeling work that has tried to incorporate fluvial erosion into numerical ice models to investigate the formation of N-channels.**

This is a very good suggestion, and it is in fact explored in detail in another manuscript in preparation. We believe that the work presented here is already extensive.

**3.1.- Morphology of the subglacial meltwater features : I believe this paper really undersells what is a fantastically high resolution study of the morphology of bedrock carved channels. I am not aware of such detailed work in such well preserved landforms and yet the results seem rather hidden away after the comparison of the different techniques. I would like to see more made (and example figures shown) of the channel morphologies, including further discussion of the headwalls, anabranching pattern, spacing, cross-sectional profiles and association with other bedforms, while**

a summary statistics table would also really help the reader. As currently written, it is the use of the different techniques which really comes out from this, not the morphology of the features.

To address this comment, we present an additional section (section 5) meant to cover in more depth the morphological characteristics of subglacial channels. We largely base our description of the channel morphology in the work by [14], and complement the field observations with additional data drawn from the elevation maps we produced. Spacing, however, will be addressed in detail in a follow-up study in preparation.

**3.2.-To broaden this work out it would have been nice to see how their dimensions compare with other studies of similar sized features (and then also the larger tunnel valleys) and to discuss what this means in terms of their formation (e.g. slow and steady vs catastrophic drainage).**

We added a comparison of our observations to other studies in table 1 [e.g., 1, 6, 13, 14], but we will not include an interpretation of channel formation mechanisms here as it will be a focus of a follow-up manuscript in preparation.

**Minor comments:**

**P1L1: Tunnel valleys can also be cut into sediment.**
We adopted the correction.

**P1L6: should be extent**
We fixed the typo.

**P2L28: I think there needs to be some recognition of the different scales here. N-channels are typically associated with much smaller channels cut specifically into bedrock. Tunnel valleys/channels may also be cut into sediment and are much large. In terms of the effect on ice dynamics most of the work is associated with the evolution to channelized drainage and these channels are again envisaged to be an order of magnitude smaller than tunnel valleys/channels.**
We added a sentence referring to the distinct scales of tunnel valleys and subglacial channels.

**P2L10: from instead of only with.**
We incorporated this suggestion.

**P2L26: e.g. in wrong position in brackets beginning Denton**
We fixed the typo.

**P2L27: See also for a comprehensive mapping along a large portion of the southern sector of the Laurentide Ice Sheet: Livingstone, S.J. and Clark, C.D., 2016. Morphological properties of tunnel valleys of the southern sector of the Laurentide Ice Sheet and implications for their formation. Earth Surface Dynamics, 4(3), p.567. Indeed, there is a large body of work in this area, and also in the North Sea: see older papers in review by Kehew et al. (2012) for completeness it would be good to reference some of the key work.**
We added this reference and a reference to an earlier review by [4] for completeness. However, we do not go in further detail about tunnel valleys in this revised manuscript, other than to compare their length scales to those of the channels on Devon Island.

**P2L34: This is conjecture where is the evidence for temporal variability and large inputs?**

We removed this sentence as it was written referring to tunnel valley identification. See however [11] for a mathematical analysis of channelization driven by melt supply variability.

**P3L2: Although there has been a large body of work on the morphology of tunnel valleys (e.g. Livingstone and Clark, 2016).**
We removed this comment. See the comment above, and major comment 1.

**P3L15: ice sheet began retreating towards the current**
We adopted this suggestion.

**P3L17: Capitalise Ice Sheet.**
We incorporated this suggestion.

**P5L11: This is not obvious from Fig. 1.**
We fixed the wording to remove the reference to fluvial vs. subglacial drainage here, to be discussed in the next section with the longitudinal profile results.

**P5L25: What about figures 2 and 3?**
Here we would refer to figure 1 and 3 or only to figure 3. We fixed this.

**P9L1-5: It would be useful for the reader if you included a schematic, perhaps on one of the profiles in Fig. 3 as it is not clear to me.**
We adopted the suggestion in figure 10, (previous figure 6) which now shows a cartoon with a representation of the metrics introduced in the text.

**P9L19: to have originated in a subglacial regime.**
We fixed the wording.

**P9L8: tributaries have widths of**
We believe there is a typo in the page/ line reference provided, it actually refers to P11L8. We fixed the typo suggested here.

**P9L8-10: I also found that apparent anabranching of the channels an interesting feature worth observing. In particular, can you tell from the DEM whether the channels were formed synchronously (same depth of channel bottom), or time-transgressively (which might manifest as different depths of anabranching channels).**
We added a subsection (subsection 5.3) about the anastomosing characteristics of some of these networks, and took a cross section profile across an anabranching section to include the point suggested by the reviewer.

**P13L1-2: More details are needed here. How does the cross-sectional shape and depth change between recognised tunnel valleys and river channels? This is a key distinction that has been glossed over here.**
We agree with the reviewer. We added a characterization of subglacial channel and river cross section (see Fig. 6), including their scales and downstream evolution, context drawn from the fluvial literature [9] for the fluvial channel cross sectional evolution in terms of the discharge in gravel streams.

**P13L3: This is not clear to me as only a small portion of the image corresponds to tributaries that you pick out as having similar widths. Is there a better example?**
We reworded the description of this figure (see comment about Fig. 5), in the context of three zones: tributary origin, tributary development, and merging into a meltwater seasonal stream.

**P13L10: Braiding is the wrong term here I believe as this would refer to temporary**

**islands as part of a dynamic sedimentary system. Anabranching is a more appropriate term**

We adopted the reviewer's comment.

**P13L8-11: Again, this seems very short on details. You state that you can pick out the key characteristics of tunnel valley networks but then seem to restrict this to a few choice observations.**

We eliminated this paragraph in the rewording of the description of this figure. In the new manuscript, section 5 offers more details on channel morphology.

**P13L13: criteria exposed before is an odd phrase. Re-write.**

We reworded this sentence.

**P13L19: delete targeted**

We deleted this word.

**P13L21: and approximately constant downstream from the origin until**

We rephrased this subsection to add more details. In particular, we added a figure describing the evolution of fluvial and subglacial cross sections, which adds more context to the following comparative discussion.

**P13L17: tunnel valley widths are up to tens**

We fixed the typo.

**Do these channels merge into the surrounding topography at their origin or do they have a clear amphitheater-headed canyons? (e.g. see Lamb et al., 2006, 2014). This might give you some clues as to their origin.**

They merge in the surrounding topography. We state this in more detail in section 5 and in the description of figure 5, as we realize it is an important aspect of the channel network morphology. In particular, in figure 5 panel (a) (photogrammetry), we show the region where all tributaries grade smoothly into the surrounding plateaus, with no topographic sign of clear heads within the resolution of the DSM (40 cm).

**P13L34: Examples of this pattern are shown in**

We fixed the typo.

**P14L1: tree-like network typical of**

We reworded this sentence to offer a more detailed description of the anabranching/ anastomosing patterns observed in the subglacial networks.

**P14L3: tunnel valleys also have very few.**

We reworded this sentence to offer a more detailed description of the anabranching/ anastomosing patterns observed in the subglacial networks. In particular, section 5.2.2 describes the number of tributaries and shape of the networks.

**P14L8-21: This is nicely summarized, but not new. The hydraulic potential gradient has been widely used to infer channel direction and we know that the ice surface slope can drive water over topographic undulations.**

We rephrased this section, and put this metric in the context of both theoretical descriptions [e.g., 10, 12, 17] and observational evidences [e.g., 8, 13, 14, 16] showing oblique subglacial channel direction to topographic gradients. We then highlight our addition, which is the quantification of this deviation (table2, column 5, figure 10) in a numerical metric.

**P15L4: And critically, their morphology and association with other subglacial fea-**

**tures like eskers, moraines, outwash fans.**
We added the list of subglacial features suggested (P20L11 in this version).

**P15L5: What is the example and how does that help? The text below does not mention other subglacial bedforms.**
We reworded this section in the context of an improved, more detailed characterization of sub-glacial channel morphology. The text in the first version of the manuscript was referring to other subglacial features as in other subglacial channels. Indeed, if two individual channels within one same network display undulations in their profile, according to criteria (4) (association with other subglacial features) we could expect the other tributaries to have formed subglacially.

**P15L10-12: I am not convinced this is a new metric for tunnel valley identification (e.g. see Greenwood et al., 2007; Livingstone and Clark, 2016 section 3.1).**
We agree with the referee. This is not a new metric, but we did suggest its quantification to improve objective and remote sensing characterization of subglacial channels.

**P15L11-12: I do not understand this final sentence. If the ice is cold based surely large meltwater channels are unlikely to form?**
Our affirmation is wrong, indeed. We removed it from the text in the revised manuscript, although in the first version we were referring to cold based ice sheets with water accumulation at the ice margin, or in a more general context where ice moves (and erodes) very slowly relative to channel formation, to avoid channel destruction.

**P16L5: How can you be so precise in stating the timing of these features?**
This affirmation was based on the assumption that the networks formed close to the ice margins, given how short the networks are (up to 2 km). However, and given the fact that we do not discuss channel formation mechanisms in the text, we removed this sentence from the conclusions.

**Figures**

**Figure 1: Can you distinguish, maybe with different colour arrows, between the river valleys and tunnel valleys. This would help the reader.**
We highlighted each network by surrounding it with a coloured box: black for fluvial and white for subglacial. We also changed the identification scheme used to refer to channel groups, an refer now to specific networks as opposed to "groups of channels", in response to a comment by M. Margold, which are also indicated in this figure.

**Figure 2: In the caption you refer to distinct groups but these are not clear from the figure. It would be useful to include these headings so the reader can easily distinguish. It is not that obvious from the profiles why some have been termed tunnel valleys and some rivers. For instance, most of group 4 and 5 tunnel valleys are relatively smooth with little in the way of reverse bed slops, and are therefore comparable to groups 2 and 3. What allowed you to distinguish these as tunnel valleys rather than river channels?**
We believe the reviewer is referring to Fig. 3 here. We changed Fig. 3 by referring directly to the channel networks introduced in the text (see comment above), and adding a title to each of these networks in the figure. We also discuss the longitudinal profiles of fluvial and subglacial systems in more detail in the text, particularly in section 5. Distinguishing subglacial channels in the basis of remote sensing data (in particular longitudinal profiles) is challenging, and it is indeed one of the objectives of this paper. Notice how we added two additional metrics that were only qualitatively discussed in the previous manuscript: a shape factor (top width to depth) and a measure of deviation from topographic gradients. The quantification of these metrics makes the distinction between both systems easier.

**Figure 4: Missing a colour legend for panels (c) and (d).**
We added the colour legend, and also a better scale for panels (a) and (b).

**Figure 5: What do the arrows refer to? More details on what is actually picked up in these images would be helpful to the reader.**
We clarified the description of figure 5, and now use this figure to describe the different stages of network evolution, from the formation of the tributaries to their merging into meltwater fed seasonal streams.

**M. Margold**

**Major comments**

**The motivation for the study and its setting within the context of existing knowledge in the field is not articulated enough. The authors might attempt to spell out more clearly what the study brings that older studies were lacking  this is a point where to refer to the existing literature on glacial meltwater channels.**
In response to M. Margold's comment, we now clearly state in the introduction how the primary motivation for this study is to provide a remote sensing based identification scheme for subglacial channels, together with the highest resolution topographic study presented regarding these drainage systems. We come back to these objectives in the discussion and conclusions.

**The methods section is lengthy and at places self-serving. Why is there a need to reproduce the surface topography at cm resolution? The authors might attempt to better align the methods used with the stated objectives.**
One of the objectives of the expedition to Devon Island (although not necessarily the primary focus of this paper) was to test the capabilities of portable LiDAR systems at analyzing channel geometries, which is the reason behind the high detail provided in the methods section. In addition, LiDAR resolution (cm scale) of the channels provides a very reliable data set for (1) analyzing the presence of inner channels in the subglacial channels, (2) providing ways to constrain the angle of channel walls (close to the angle of repose), and (3) ground truthing the GPS filtered longitudinal profile data in figure 3. We acknowledge, however, the disproportionately large methods section compared to results in the previous version of the manuscript, and we have addressed this issue by adding section 5 with a much more detailed geomorphological analysis of subglacial channels. This section includes cross sectional characteristics and evolution, network directionality with respect to regional slopes, characteristics of the channel heads, etc. , and addresses the need to better align methods, objectives, and results.

**The distinction between tunnel valley erosional regime and river valley erosional regime is vague. Ideally, the authors might qualify the main characteristics of subglacial and fluvial drainage (based on literature) and look for the characteristic features in their data. The manuscript is overly relying on Kehew et al. (2012).**
We acknowledge our mistake in identifying our features as tunnel valleys. Indeed, as M. Margold and S. Livingstone point out, these channels are better described as subglacial channels in terms of their spatial scales. To fix this issue, we removed most of the references to the work by [7] as it does not apply to subglacial channels, and added relevant literature [e.g., 1, 6, 8, 14, 16]. Addressing the first part of the comment, in this revised version we give further references to fluvial erosion regimes: i.e., the description of bankfull width in gravel rivers by [9] or the introduction of a shape factor (width over depth, widely used in the fluvial literature), which helps to quantify the characteristics of fluvial vs. subglacial cross sections.

**Portions of the text that refer to the figures read very much like figure captions.**
We agree with the reviewer, particularly regarding the photogrammetry and LiDAR figures (Fig.

4 and Fig. 5). We fixed this issue by avoiding repetition with figure captions, by adding further descriptions and implications of the figures, and by removing the text that did not add any additional information from the caption.

**Broader, v-shaped cross profile of the river valleys vs. narrower, flat-bottom, steep-walled cross profile of the meltwater channels could something be inferred about the discharge and the length of formation/operation of the feature(s)? While this goes beyond the scope of the manuscript, a few references could be provided where this topic might be followed.**
This is indeed a topic of interest and will be analysed in further detail in a second manuscript in preparation, particularly regarding the relationship between spacing and discharge [e.g., 2, 17]. In this study we introduced the shape factor to capture quantitatively some of these differences, in particular the top width evolution of fluvial vs. subglacial channels. To our knowledge, however, hydraulic relationships evaluating discharge vs. bankfull width for subglacial channel cross sections have not been derived, and in fact there is little work addressing the mechanics of erosion in N channels (see i.e., [2, 5, 17]). Erosion efficiency is often calculated per unit channel width, which makes the estimation of such relationship difficult (algebraically, it is possible to derive one from the work by [15] for canals, but it does not apply to the channels here described).

**Minor comments:**

**P1 L8 Kinematic mobile LiDAR. I am not an expert on this instrumentation but from checking briefly online, either one or the other adjective is usually used. Pairing the two adjectives seems to make little sense to me since they mean largely the same, just one having a Greek root and the other a Latin one.**
We fixed this mistake and refer to it as Kinematic LiDAR Scan (KLS) as in other sections of the text. Our intention is to refer to its portability (hence mobile), which is the asset of this new technique.

**P2 L23 Younger Dryas**
We fixed the typo.

**P2 L33 Criteria is plural, write criterion where it is a singular.**
We fixed this issue throughout the manuscript.

**P2 L34 Warm-based is a more common term than wet-based**
We adopted the reviewer's suggestion.

**P3 L14-17 Ages in Dyke (1999) are in radiocarbon years, however, the notation ka BP is now commonly used for calendar years. Either state that it is C-14 years or calibrate.**
We adopted the reviewer's suggestion.

**P3 L20 remove the full stop before the reference**
We fixed the typo.

**P3 L34 deposition landforms is a more common term**
We adopted the reviewer's suggestion.

**P5 L12-13 Downstream of. . . I dont understand what do you mean with this sentence.**
We reworded this sentence by referring to stream orders: once the network develops a stream order of 2 or more.

**P8 L24-25 Check the wording, represent appears two times**
We fixed the typo.

**P8 L32-33 requires quantitative field longitudinal profile observations reword to requires measuring longitudinal profiles in the field.**
We adopted the reviewer's suggestion.

**P9 L24 agreement**
We fixed the typo.

**P13 L3 replace in panel (b) with in Fig. 5b**
We adopted the reviewer's suggestion.

**P13 L4 replace packs with accumulations**
We adopted the reviewer's suggestion.

**P13 L13 criteria exposed before exposed does not work here very well, search for a more fitting verb (stated, listed).**
We reworded this sentence to make it specific to the longitudinal profile analysis.

**P13 L31 shallow**
We fixed the typo

**P13 L33-34 This is something that should be discussed further with references to older literature.**
We reworded this whole section when we added the more detailed characterization of subglacial channel morphology. We do provide a description of channel and network shape in section 5 that includes references to older literature, in particular the observations by [1, 14].

**P14 L8-10 We argue here that differences among channel direction and local topographic gradients are also indicative of subglacial erosion in areas where the ice erosion rate by sliding is lower than the meltwater erosion rate (Weertman, 1972; Paterson,1994). These can be submarginal meltwater channels that record the ice surface slope direction but do not necessarily bear any evidence with regard to ice erosion rate.**
The previous draft was missing any reference to or discussion about lateral meltwater channels, and therefore it was impossible for the reader to make this distinction. Following a comment by M. Margold (see above in Major Comments) we incorporated a full subsection describing the differences between lateral and subglacial channels, and showed how the networks analysed are subglacial. The reference about fast/ slow sliding rates, and the destruction of N-channels that do not follow ice flow lines in fast-moving glaciers and ice sheets is old [17]. Given our distinction between lateral and subglacial meltwater channels, we believe we have addressed this comment with no further changes needed.

**P15 L1-4 Here the fact that you have been ignoring all the types of meltwater channels other than subglacial really becomes problematic because you might be dealing with lateral or submarginal channels in this case.**
We fixed this issue (see above in Major Comments).

**15 L10-12 Again, there is a possibility that these might be submarginal or lateral meltwater channels. It might well be that most or all the channels that you classify as subglacial indeed are subglacial and not submarginal or lateral. But you need to be provide argumentation for this.**
see comment above.

**Figures**

**Fig. 3 Add group numbers or panel letters so that the groups can be easily identified.**
We adopted the reviewer's suggestion

**The figure would be more informative if one could see the topographic settings of the pictured groups of channels. Could the photographs possibly be draped on a DEM derived hillshade?**
We adopted the reviewer's suggestion in an additional figure (Fig. 6), which we also use to identify lateral meltwater channels from subglacial channels.

**References**

[1] Beaney, C. L., and J. Shaw, The subglacial geomorphology of southeast alberta: evidence for subglacial meltwater erosion, *Canadian Journal of Earth Sciences*, *37*(1), 51–61, 2000.

[2] Beaud, F., G. E. Flowers, and J. G. Venditti, Efficacy of bedrock erosion by subglacial water flow, *Earth Surface Dynamics*, *4*(1), 125–145, 2016.

[3] Brennand, T. A., Macroforms, large bedforms and rhythmic sedimentary sequences in subglacial eskers, south-central ontario: implications for esker genesis and meltwater regime, *Sedimentary Geology*, *91*(1-4), 9–55, 1994.

[4] Cofaigh, C. Ó., Tunnel valley genesis, *Progress in Physical Geography*, *20*(1), 1–19, 1996.

[5] Creyts, T. T., G. K. Clarke, and M. Church, Evolution of subglacial overdeepenings in response to sediment redistribution and glaciohydraulic supercooling, *Journal of Geophysical Research: Earth Surface*, *118*(2), 423–446, 2013.

[6] Greenwood, S. L., C. D. Clark, and A. L. Hughes, Formalising an inversion methodology for reconstructing ice-sheet retreat patterns from meltwater channels: application to the british ice sheet, *Journal of Quaternary Science*, *22*(6), 637–645, 2007.

[7] Kehew, A. E., J. A. Piotrowski, and F. Jørgensen, Tunnel valleys: Concepts and controversiesa review, *Earth-Science Reviews*, *113*(1), 33–58, 2012.

[8] Livingstone, S. J., W. Chu, J. C. Ely, and J. Kingslake, Paleofluvial and subglacial channel networks beneath humboldt glacier, greenland, *Geology*, *45*(6), 551–554, 2017.

[9] Parker, G., P. R. Wilcock, C. Paola, W. E. Dietrich, and J. Pitlick, Physical basis for quasi-universal relations describing bankfull hydraulic geometry of single-thread gravel bed rivers, *Journal of Geophysical Research: Earth Surface*, *112*(F4), 2007.

[10] Paterson, W., *The physics of glaciers*, Butterworth-Heinemann, 1994.

[11] Schoof, C., Ice-sheet acceleration driven by melt supply variability, *Nature*, *468*(7325), 803–806, 2010.

[12] Shreve, R., Movement of water in glaciers, *Journal of Glaciology*, *11*(62), 205–214, 1972.

[13] Sissons, J., A subglacial drainage system by the tinto hills, lanarkshire, *Transactions of the Edinburgh Geological Society*, *18*(2), 175–193, 1961.

[14] Sugden, D. E., G. H. Denton, and D. R. Marchant, Subglacial meltwater channel systems and ice sheet overriding, asgard range, antarctica, *Geografiska Annaler. Series A. Physical Geography*, pp. 109–121, 1991.

[15] Walder, J., and A. Fowler, Channelized subglacial drainage over a deformable bed, *Journal of Glaciology*, *40*(134), 1994.

[16] Walder, J., and B. Hallet, Geometry of former subglacial water channels and cavities, *Journal of Glaciology*, *23*(89), 335–346, 1979.

[17] Weertman, J., General theory of water flow at the base of a glacier or ice sheet, *Reviews of Geophysics*, *10*(1), 287–333, 1972.

---

## Author Response (AR2)

**Response to the Editor's Comments**

Anna Grau Galofre A. Mark Jellinek Gordon R. Osinski Michael Zanetti Antero Kukko

March 7, 2018

We would like to thank the editor for considering the revised version of this manuscript, and for this prompt revision. In response to his comments, we corrected the typo he pointed out, located and corrected two other typos, and added a summary of the key observations presented in section 5 to the conclusions of the manuscript. We tracked these changes in the manuscript and present them in the rest of this document.

Sincerely,

[revised manuscript text omitted]